# Peripheral sensory coding through oscillatory synchrony in weakly electric fish

Christa A Baker, Kevin R Huck, Bruce A Carlson*

Department of Biology, Washington University in St. Louis, St. Louis, United States

**Abstract** Adaptations to an organism's environment often involve sensory system modifications. In this study, we address how evolutionary divergence in sensory perception relates to the physiological coding of stimuli. Mormyrid fishes that can detect subtle variations in electric communication signals encode signal waveform into spike-timing differences between sensory receptors. In contrast, the receptors of species insensitive to waveform variation produce spontaneously oscillating potentials. We found that oscillating receptors respond to electric pulses by resetting their phase, resulting in transient synchrony among receptors that encodes signal timing and location, but not waveform. These receptors were most sensitive to frequencies found only in the collective signals of groups of conspecifics, and this was correlated with increased behavioral responses to these frequencies. Thus, different perceptual capabilities correspond to different receptor physiologies. We hypothesize that these divergent mechanisms represent adaptations for different social environments. Our findings provide the first evidence for sensory coding through oscillatory synchrony.

## Introduction

Evolutionary change in sensory systems can have profound effects on perception and behavior (*Carlson and Arnegard, 2011*; *Carlson, 2012*; *Baldwin et al., 2014*). Recently, we documented a striking example of evolutionary change in the electrosensory systems of weakly electric fish (*Carlson et al., 2011*). Anatomical modifications of sensory receptors and the associated brain pathway that processes electric communication signals resulted in a newfound ability to detect subtle variations in these signals. This perceptual expansion led to a dramatic increase in the rates of signal evolution and species diversification. However, how these anatomical and perceptual differences relate to physiological differences in the underlying information processing mechanisms was unknown. Here, we investigate how evolutionary divergence in sensory perception relates to differences in peripheral sensory coding.

Mormyrid fish communicate by producing a species-specific electric organ discharge (EOD) at variable interpulse intervals (IPIs) (see *Carlson, 2002* for review). The ability of some species to detect subtle variation in EOD waveforms is related to differences in the anatomy and physiology of peripheral sensory receptors called knollenorgans (*Harder, 1968a*, *1968b*; *Carlson et al., 2011*). In species sensitive to waveform variation, these receptors are broadly distributed across the head, back, and underbelly (*Carlson et al., 2011*). These receptors fire a single, time-locked spike in response to an electric stimulus in all species previously studied (*Bennett, 1965*; *Harder, 1968b*; *Hopkins and Bass, 1981*; *Lyons-Warren et al., 2012*). Spiking receptors encode electric pulse duration into spike-timing differences, with some receptors responding to pulse onset and others to pulse offset (*Hopkins and Bass, 1981*). These timing differences are compared in the midbrain (*Friedman and Hopkins, 1998*; *Lyons-Warren et al., 2013*).

*For correspondence: carlson. bruce@wustl.edu

Competing interests: The authors declare that no competing interests exist.

**eLife digest** The mormyrids are a family of fish that can generate and detect weak electric fields that they use to navigate and communicate. Each species in this family produces its own distinct shape of electrical signals. Other fish detect these signals using structures called receptors, which then send information to the brain in the form of an electrical nerve impulse.

Previous research showed that different species of fish respond to different aspects of the electric fields they detect, depending on the receptor types that they have evolved. For example, some species have 'spiking receptors' that send a spike of activity to the brain at the start or end of a detected electrical pulse. This allows the fish to detect subtle changes to the shape of the detected electric field. Other species have sensors known as oscillatory receptors; these send a continuous wave of nerve activity to the brain even when no electric field is detected. It is not clear exactly how oscillatory receptors work or what the roles of these different receptor types are.

Baker et al. have now recorded the activity of the receptors of mormyrid fish as they were exposed to electrical fields that mimic the signals they are normally exposed to in their daily environment. The recordings confirmed previous results that suggest that species with spiking receptors are able to detect variations in the shapes of electrical signals produced by other fish because the receptors produce spikes of activity at the start or end of a detected signal. This response pattern allows their brains to analyze the signal shape.

In contrast, Baker et al. found that when oscillatory receptors detect a new electric field, they reset their pattern of nerve activity, causing multiple receptors to briefly synchronize their activity. This enables the fish to detect where a signal has been sent from. However, the oscillatory receptors are unable to detect any variations in the shape of the detected electric signal. The oscillatory receptors are most sensitive to certain electric field patterns that are produced by large groups of a single species of fish. Further experiments showed that the fish also change their behavior when these particular electric field patterns are detected.

Why some species of mormyrids have spiking receptors while others have oscillatory receptors is unknown. It is also unknown how the brain processes oscillatory receptor activity. Future studies will investigate whether these two receptor types may be linked to differences in fish's social behavior, and how the neural networks for processing sensory information differ between the two types of fishes.

In contrast, species insensitive to waveform variation have receptors that produce spontaneously oscillating potentials at frequencies up to 3 kHz (*Harder, 1968b*). These receptors are clustered into three groups, called rosettes, on each side of the head (*Harder, 1968b*; *Lavoue et al., 2004*, *2010*; *Carlson et al., 2011*). How oscillatory receptors encode electric communication signals is unknown.

Oscillations in neural activity are widespread throughout the brains of many animals, including humans, and are known to play important roles in generating rhythmic motor behaviors (*Ramirez et al., 2004*; *McCrea and Rybak, 2008*). However, how oscillations contribute to processes such as sleep, attention, memory, motor output, and sensory coding remains unclear (*Buzsaki and Draguhn, 2004*; *Thut et al., 2012*; *Canavier, 2015*). Oscillations could facilitate multiplexed signal coding in which spikes carry information depending on the phase of the oscillation during which they occur (*Thut et al., 2012*). Another effect of oscillations could be to enhance communication between neural populations during periods of synchronized excitability (*Thut et al., 2012*). Additional hypothesized functions include signal gating, feature binding, and cross-modal integration (*Thut et al., 2012*), although a clear functional role for electrical oscillations in sensory processing has remained elusive (*Canavier, 2015*).

To gain insight into the neural basis of differences in perceptual abilities among species, here we investigate for the first time how oscillating receptors encode electric stimuli. Using extracellular recordings, we reveal a novel phase-reset mechanism that mediates signal detection by transiently synchronizing the oscillations among receptors. This is the first demonstration of information coding by oscillatory synchrony at the periphery. These phase resets did not encode pulse waveform, explaining why species with oscillating receptors cannot behaviorally discriminate EOD waveform variation. Furthermore, we provide physiological and behavioral evidence that oscillating receptors

respond most strongly to communication signals with fast temporal patterns that are only produced by large groups of conspecifics, suggesting that oscillating receptors may be specialized for group signal detection.

## Results

### Spontaneous receptor activity

To measure baseline activity rates, we recorded extracellular spontaneous activity from the sensory receptors of five species (*Figure 1*). This sample included two species with broadly distributed spiking receptors, as well as *Petrocephalus microphthalmus*, whose receptor physiology was previously unknown. *P. microphthalmus* is behaviorally sensitive to EOD waveform variation, but it evolved this ability independently of other species with broadly distributed receptors (*Carlson et al., 2011*). We found that the receptors of *P. microphthalmus* indeed fire spikes (*Figure 1A*), just like other mormyrids with broadly distributed receptors and EOD waveform sensitivity. We also recorded spontaneous oscillations from rosette receptors of two species (*Figure 1C,D*). Spontaneous interspike intervals and spontaneous oscillation periods varied across species (*Figure 1B,D*).

Oscillating receptors are organized into three rosettes on each side of the head (*Figure 2A*) (*Harder, 1968b*; *Lavoue et al., 2004*, *2010*; *Carlson et al., 2011*). To study how spontaneous activity varied within a rosette, we obtained spontaneous recordings from every receptor in a single rosette in one *Petrocephalus tenuicauda* (*Figure 2B*). Spontaneous oscillation amplitude was largest in the center of the rosette and progressively decreased towards the periphery. Oscillation frequency and amplitude were not significantly correlated (Spearman R = 0.27, $t_{(22)}$ = 1.3, p = 0.21), although frequencies tended to be highest near the center of the rosette (*Figure 2B*).

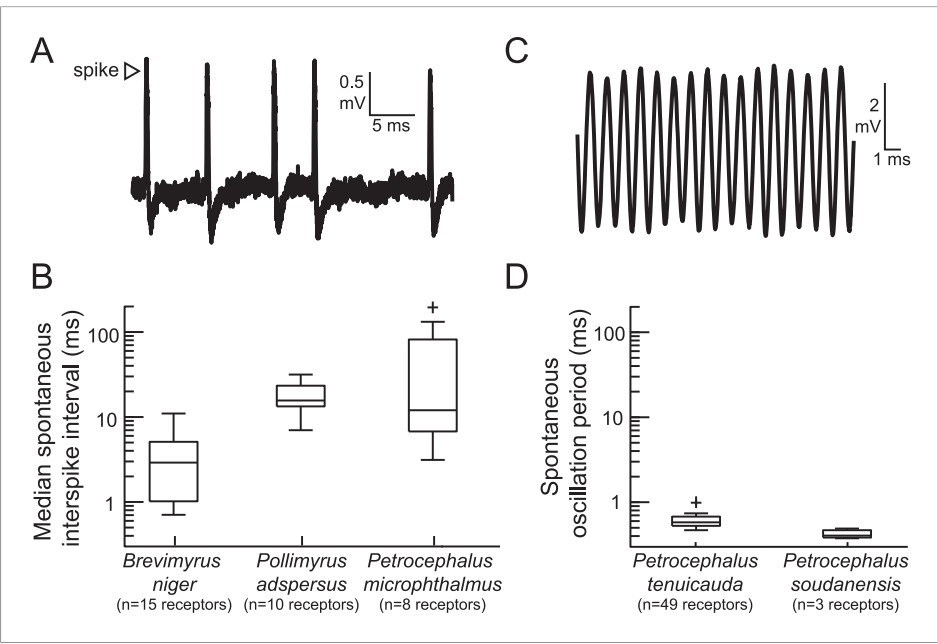

**Figure 1**. The peripheral sensory receptors of some mormyrid species fire spikes, whereas the receptors of other species produce spontaneously oscillating potentials. (**A**) An extracellular recording of spontaneous spikes from a receptor of *Petrocephalus microphthalmus*. (**B**) Box plots of spontaneous interspike intervals in the receptors of three species. (**C**) An extracellular recording of spontaneously oscillating receptor potentials in *Petrocephalus tenuicauda*. (**D**) Box plots of spontaneous oscillation periods in the receptors of two species. In *P. tenuicauda*, the amplitude of oscillatory activity varied from 0.04 to 6.1 mV. We only measured the frequency of spontaneous activity if the oscillation amplitude was at least 2.5 times baseline noise (≥0.1 mV). 49 of 69 (71%) receptors met these criteria. The receptors of *Petrocephalus soudanensis* had spontaneous oscillation amplitudes of 0.09, 0.4, and 1.9 mV and frequencies of 2.1, 2.6, and 2.5 kHz, respectively.

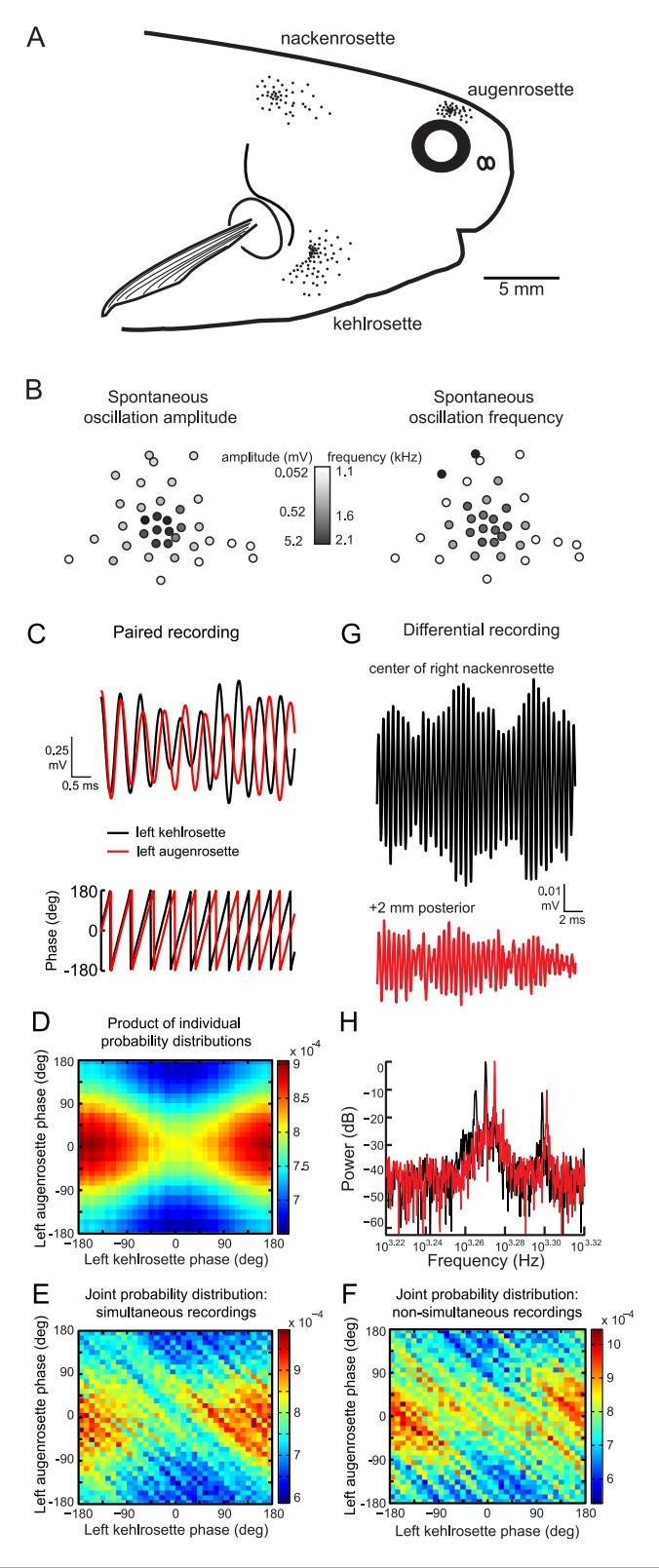

**Figure 2**. Spontaneous oscillatory activity is not synchronized across receptors. (**A**) Receptor locations on the right side of one *P. tenuicauda* are illustrated as black dots (receptor size not to scale). (**B**) An approximate map of all 36 receptors in the right augenrosette of one *P. tenuicauda*. This map comes from a different fish than shown in **A**. (**C**) Top, simultaneous extracellular recordings from a receptor in the left kehlrosette (black) and a receptor in the left

*Figure 2. continued on next page*

*Figure 2. Continued*

augenrosette of one *P. tenuicauda*. Bottom, instantaneous oscillatory phases of the simultaneous recordings obtained through Hilbert transform of the recorded potentials. (**D**) The product of the individual probability distributions of instantaneous phases from the same two receptors shown in **C**. The product of the individual probability distributions from each of five separate 1-s recordings was calculated. The average product across these five recordings is shown. (**E**) The joint probability distribution of instantaneous phases of the two receptors shown in **C** and **D**. The joint probability distribution was calculated over five separate 1-s recordings and then averaged. (**F**) The joint probability distribution of instantaneous phases of the first of five recordings from one receptor and the last of five recordings from the other receptor, and vice versa. The joint probability distributions from these two non-simultaneous recording pairs were averaged. (**G**) Differential extracellular recordings from a position centered over the right nackenrosette (black) and a position 2 mm posterior (red) in one *P. tenuicauda*. The recording and reference terminals of the electrode were separated by 5 mm. (**H**) Power spectra for the differential recordings shown in **G**. Only the frequency range where peaks occurred is shown (∼1.7–2.1 kHz).

## Spontaneous oscillations of different receptors are not synchronized

To determine whether the oscillations of receptors in different rosettes were synchronized with one another, we collected simultaneous recordings from pairs of receptors, each located in a different rosette (see *Figure 2C*, top). We then determined the phase of the oscillations throughout each pair-wise recording using a Hilbert transform (see *Figure 2C*, bottom). Next, we constructed probability distributions of the phases for each receptor in pair-wise recordings. If the phases are truly independent, then the joint probability histogram should be similar to the product of the individual probability distributions. Indeed, the joint probability histograms (e.g., *Figure 2E*) closely resembled the product of the individual probability distributions (e.g., *Figure 2D*). We also computed the joint probability histogram for surrogate data consisting of non-simultaneously recorded oscillations (e.g., *Figure 2F*). These histograms should match those of the simultaneous recordings if the oscillations are truly independent. Indeed, the joint probability distributions of non-simultaneous recordings closely resembled those of the simultaneous recordings (compare *Figure 2F* with *Figure 2E*).

To evaluate statistically the possibility of phase coupling between receptors in different rosettes, we determined the circular correlation coefficient ($r$) of the instantaneous oscillatory phases between each receptor pair (*Berens, 2009*). We averaged the resulting $r$ over five 1-s recordings from each pair. Across 14 receptor pairs in one *P. tenuicauda*, the mean $r$ was $0.05 \pm 0.05$. For comparison, we generated non-simultaneously recorded surrogate data sets consisting of the first of five 1-s recordings from one receptor in the pair and the fifth recording from the other pair, and vice versa. There was no significant difference between the $r$ of simultaneous recordings and the $r$ (mean = $4 \times 10^{-4} \pm 7 \times 10^{-4}$) of non-simultaneous recordings (Wilcoxon matched-pairs test, $Z_{(14)} = 1.7$, p = 0.084). In another *P. tenuicauda*, we recorded from six pairs of receptors and found similar results, with no differences in the circular correlation coefficient for phases of simultaneous (mean $r = -0.06 \pm 0.04$) and non-simultaneous (mean $r = 0.001 \pm 0.001$) recordings (Wilcoxon matched-pairs test, $Z_{(6)} = 1.8$, p = 0.075). Therefore, oscillatory phase was independent across receptors in different rosettes.

To determine whether the individual receptors within rosettes oscillated at the same frequency, we recorded spontaneous activity with a differential electrode pair centered over a single rosette. The differential electrode allowed us to record local field potentials resulting from the combined oscillations of multiple receptors. If the receptors near our electrodes were oscillating independently at different frequencies, we should see beats, or amplitude modulations, in the field potentials due to constructive and destructive interference. Indeed, beats were present in these recordings (see example in *Figure 2G*), indicating that our electrode was in fact recording activity from multiple sources with different frequencies. Accordingly, at least three distinct peaks were present in the power spectrum of this recording (black trace in *Figure 2H*). Recording activity at a position 2 mm posterior to this location led to the loss of two of these peaks and the addition of two new peaks, suggesting that the electrode was recording activity from a distinct, but overlapping population of receptors compared to the previous location (red traces in *Figure 2G,H*). Qualitatively similar results were seen in differential recordings from additional rosettes. Therefore, in agreement with a previous report (*Harder, 1968b*), oscillatory activity of individual receptors is not correlated within or across rosettes.

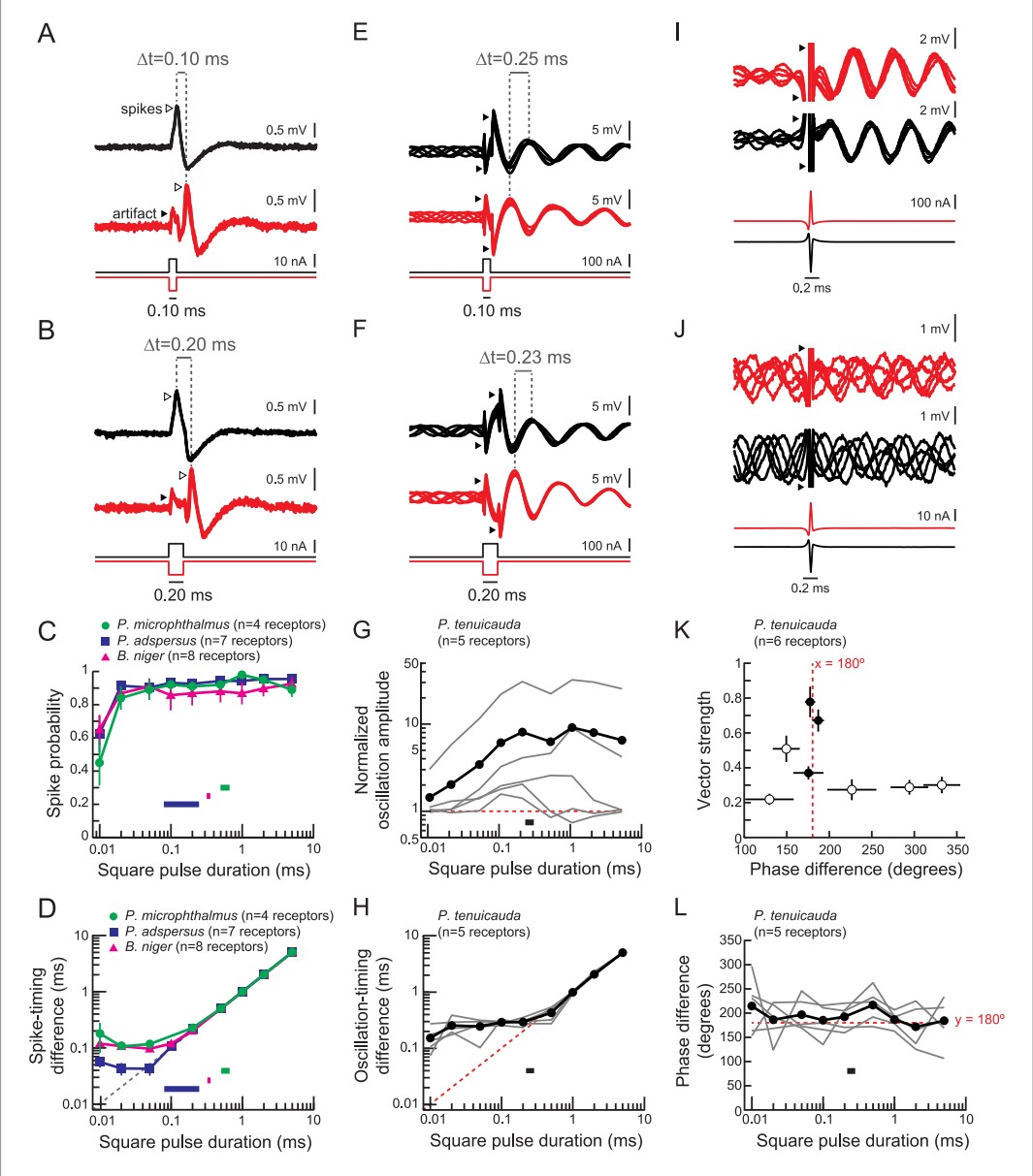

**Figure 3**. Spiking receptors encode pulse duration, whereas oscillating receptors do not. (**A**, **B**) Extracellular recordings from a spiking receptor in *B. niger* during stimulation with positive-polarity (black) and negative-polarity (red) monopolar square pulses of 0.10-ms duration (**A**) and 0.2-ms duration (**B**). In both cases, the difference in spike times (Δt) matches the pulse duration. Traces from five stimulus repetitions are superimposed. (**C**) Spike probability vs square pulse duration for responses to positive-polarity stimuli for spiking receptors of three species. Horizontal bars indicate the behaviorally relevant ranges of total durations measured in 10 conspecific electric organ discharge (EOD) waveforms. (**D**) Spike-timing differences between responses to positive- and negative-polarity stimuli vs square pulse duration for the same receptors in **C**. (**E**, **F**) Extracellular recordings from an oscillating receptor in *P. tenuicauda* during stimulation with positive- (black) and negative- (red) polarity monopolar square pulses of 0.10-ms duration (**E**) and 0.20-ms duration (**F**). In both cases, the timing difference (Δt) between oscillatory peaks elicited by opposite-polarity stimuli does not match the pulse duration. Traces from five stimulus repetitions are superimposed. (**G**) Evoked oscillation amplitudes normalized to prestimulus oscillation amplitudes vs square pulse duration for oscillating receptors. Curves for each receptor are shown in gray, and the averages across receptors are shown in black. The horizontal bar indicates the range of total durations of 10 conspecific EODs. (**H**) Oscillation-timing differences between responses to positive- and negative-polarity stimuli vs square pulse duration for the same receptors shown in **G**. (**I**, **J**) Extracellular recordings from an oscillating receptor in *P. tenuicauda* in response to a head-positive ('normal polarity') conspecific EOD waveform (black) and the reverse-polarity waveform (red) at an

*Figure 3. continued on next page*

*Figure 3. Continued*

intensity of 316 nA (**I**) and 32 nA (**J**). Traces from five stimulus repetitions are superimposed. (**K**) Vector strength vs phase difference of oscillatory responses to opposite-polarity conspecific EODs in *P. tenuicauda*. Each point represents the mean of responses from six receptors. Vector strength was averaged across stimulus polarity within each receptor before averaging across receptors. Error bars represent S.E.M. Closed circles indicate phase resets that were significantly different for normal- vs reversed-polarity EODs (Hotelling test for paired circular data, $F > F_{crit}$ = 6.9, p < 0.05). The stimulus intensities that evoked significantly different phase resets for opposite polarity EODs were 56, 178, and 316 nA. Intensities of 100 nA and <56 nA did not result in significantly different phase resets. (**L**) Phase differences between responses to positive- and negative-polarity stimuli vs square pulse duration for oscillatory responses to monopolar square electric pulses.

## Electric pulses elicit time-locked spikes in spiking receptors and phase resets in oscillating receptors

To understand sensory encoding by these receptors, we recorded responses to focal stimulation with monopolar square pulses (*Figure 3*). We delivered both positive- and negative-polarity pulses to mimic the stimulation that receptors at different locations on the fish's body would experience during a natural global stimulus (*Hopkins and Bass, 1981*). As shown previously, spiking receptors responded with a single time-locked spike in response to inward current transients (*Bennett, 1965*), which occur at the onset of a positive-polarity pulse and at the offset of a negative-polarity pulse (*Figure 3A,B*). Spike probability depended on pulse duration (*Figure 3C*) but was high for stimulus durations within the range of total durations of conspecific EODs (horizontal bars in *Figure 3C*).

Oscillating receptors responded to square pulses with phase resets and amplitude increases (*Figure 3E,F*), which occurred both in receptors with high-amplitude spontaneous oscillations as well as in receptors with little to no spontaneous oscillatory activity. The degree of amplitude enhancement relative to prestimulus amplitudes depended on pulse duration (*Figure 3G*). At the stimulus duration that elicited the maximum oscillation amplitude in each receptor, there was a positive correlation between spontaneous and stimulus-evoked oscillation amplitudes (Spearman R = 0.90, $t_{(3)}$ = 3.6, p = 0.037).

## Spiking receptors encode pulse duration, whereas oscillating receptors do not

Spiking receptors encode square pulse duration into differences in spike times between receptors (*Hopkins and Bass, 1981*; *Lyons-Warren et al., 2012*). For instance, the differences in spike times elicited by each polarity of a square pulse of durations 0.10 and 0.20 ms were exactly 0.10 and 0.20 ms, respectively (*Figure 3A,B*). Across all spiking receptors, this spike-timing difference perfectly matched pulse duration for pulses longer than ∼0.1 ms (*Figure 3D*). Importantly, spike-timing differences accurately represented pulse duration over the behaviorally relevant ranges for total EOD duration in each species (horizontal bars in *Figure 3D*). Such precise coding of stimulus-timing cues mediates the demonstrated ability of species with spiking receptors to detect EOD waveform variation (*Hopkins and Bass, 1981*; *Arnegard et al., 2006*; *Machnik and Kramer, 2008*; *Feulner et al., 2009*; *Carlson et al., 2011*).

In oscillatory receptors, positive- and negative-polarity square pulses elicited phase resets that differed by ∼180° (*Figure 3E,F*). This phase difference was constant regardless of pulse duration (*Figure 3L*). To determine whether oscillating receptors could encode pulse duration in a manner similar to the spike-timing differences of spiking receptors, we measured the difference in times of oscillatory peaks evoked by opposite-polarity pulses (*Figure 3E,F*). For pulse durations exceeding receptors' intrinsic oscillation periods (∼0.5 ms), the phase could be reset in response to each stimulus edge. In these instances, we measured the oscillation-timing difference elicited by the onset of a positive-polarity pulse and the offset of a negative-polarity pulse. However, pulses of shorter durations did not elicit phase resets to each stimulus edge, since these durations are shorter than receptors' intrinsic oscillation periods. For these stimulus durations, we measured the oscillation-timing differences after stimulus offset (e.g., *Figure 3E,F*). Oscillation-timing differences did not accurately encode pulse durations shorter than ∼1 ms (*Figure 3H*), reflecting phase resets to the

leading edge of the pulse that differ by ~180° regardless of pulse duration (*Figure 3E,F,L*). This means that oscillatory responses could not encode pulse durations found in conspecific EODs (black bar in *Figure 3H*). Therefore, species with oscillating receptors cannot behaviorally detect waveform variation at least in part because the precise timing cues are not encoded at the periphery. However, oscillating receptors would be able to encode the location of the stimulus based on which receptors reset to a peak first, owing to differences in stimulus polarity.

To confirm that natural EOD waveforms also elicit oscillatory phase differences of ~180°, we presented conspecific EODs at varying intensities to oscillating receptors (*Figure 3I,J*). We measured the vector strength and phase differences for responses to opposite-polarity stimuli. The vector strength is a measure of phase-locking that ranges from 0 (phase is random across repetitions) to 1 (identical phase across repetitions). At each intensity, we compared the phases elicited by each polarity using the parametric second-order Hotelling test for paired circular data (*Zar, 1999*). At intensities for which the phase resets evoked by opposite polarity EODs were significantly different, phase differences were always ~180° (filled circles in *Figure 3K*).

## Frequency sensitivity of spiking receptors matches the power spectra of conspecific EOD waveforms

Spiking receptors have been reported to be most sensitive to the frequencies occurring in conspecific EOD waveforms (*Hopkins, 1981*; *Bass and Hopkins, 1984*; *Lyons-Warren et al., 2012*). To confirm this finding in our test species, we compared the frequency sensitivity of individual spiking receptors to the power spectra of conspecific EODs (*Figure 4A,B,D*). We collected threshold frequency tuning curves as described previously (*Lyons-Warren et al., 2012*). We defined a receptor's best frequency as the frequency with the lowest threshold. In general, spiking receptors had best frequencies that were close to the frequencies with the highest power in conspecific EODs (compare *Figure 4D* with *Figure 4B*). On average, the best frequencies of spiking receptors were within one octave of the peak power frequencies of conspecific EODs (filled symbols in *Figure 4F*).

## Frequency sensitivity of oscillating receptors does not match the power spectra of conspecific EOD waveforms

To determine the frequency sensitivity of oscillating receptors, we also collected frequency tuning curves from oscillating receptors. Due to the oscillatory nature of both stimulus artifact and response, we could not use the same constant-frequency stimuli that we used to characterize frequency tuning in spiking receptors. Instead, we presented single-cycle bipolar sine waves at three intensities and measured the vector strength of phase resets at stimulus offset. We defined an oscillating receptor's best frequency as the frequency that elicited the largest vector strength.

Oscillating receptors were most sensitive to frequencies well below the peak power frequency in conspecific EODs (compare *Figure 4E* with black trace in *Figure 4C*). On average, the best frequencies of oscillating receptors were more than two octaves below the peak power frequencies of conspecific EODs (open symbols in *Figure 4F*).

At the stimulus intensity where tuning was sharpest (10 nA), the best frequencies (0.8–2.5 kHz) of oscillating receptors roughly corresponded to their intrinsic oscillation frequencies (1.5–2.0 kHz), suggesting that their frequency tuning resulted from resonance with their spontaneous oscillations. To test this hypothesis, we also delivered single-cycle bipolar sine waves of durations equal to multiples of each receptor's intrinsic oscillation period. Oscillatory phase-locking was strongest for stimulus durations matching each receptor's spontaneous oscillation period (*Figure 4G*).

## Spiking receptors encode IPIs into interspike intervals

Electric communication signals consist of the EOD produced at variable IPIs. Whereas the stereotyped EOD waveform can contain identifying information, such as species and sex, the IPIs convey behavioral state (see *Carlson, 2002* for review). To investigate IPI coding in the three species with spiking receptors included in this study, we presented receptors with a pair of 0.2-ms duration monopolar square pulses at a range of IPIs (*Figure 5A,B*).

The probability that a receptor fired a spike in response to both pulses in the pair decreased at the shortest intervals tested (*Figure 5B,C*). However, receptors of all species reliably fired spikes at the reported minimum IPIs produced by individual fish (~8 ms) (*Hopkins, 1986*; *Carlson, 2002*).

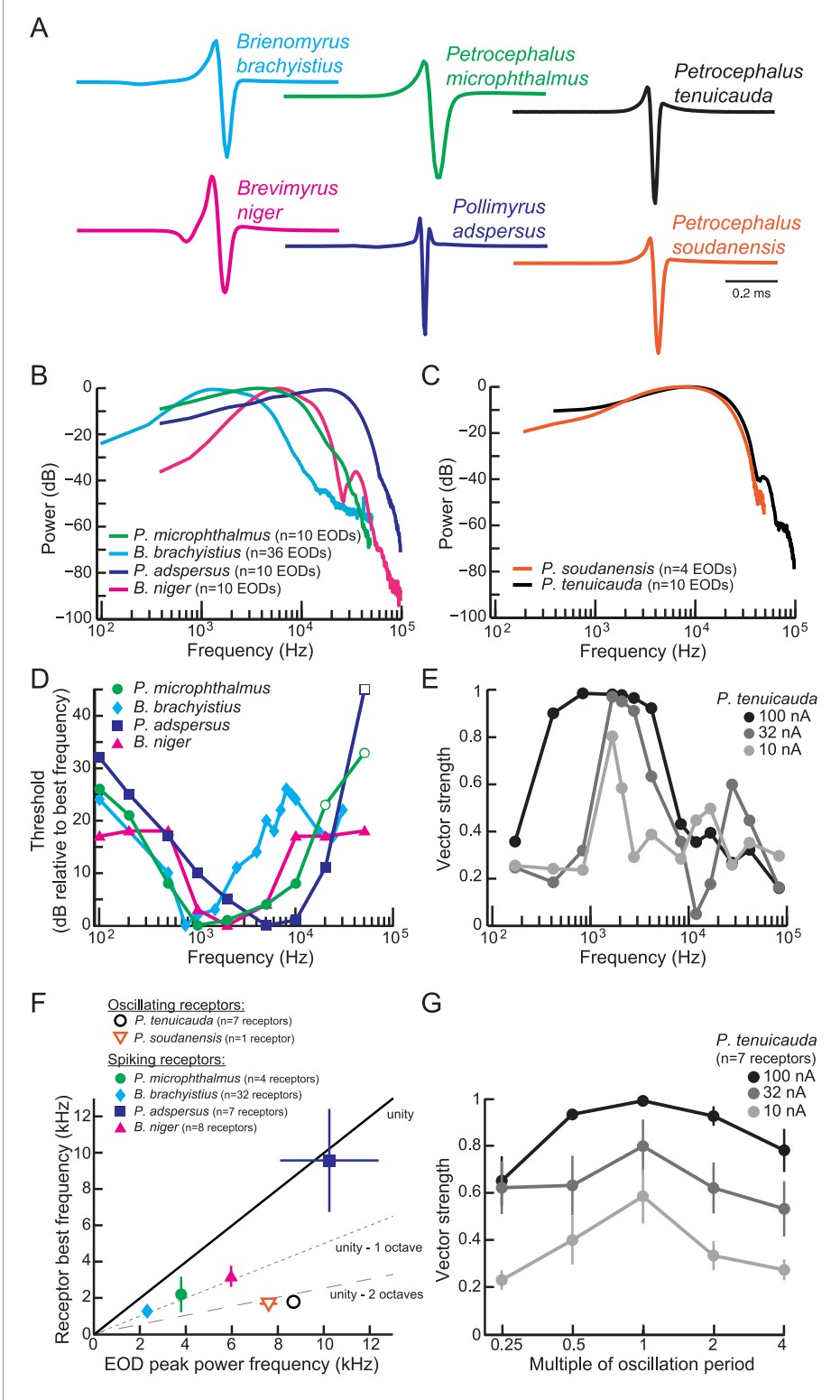

**Figure 4**. Frequency sensitivity of spiking receptors is matched to conspecific EOD power spectra, whereas frequency sensitivity of oscillating receptors is not. (**A**) Representative EODs from four species with spiking receptors (*Brienomyrus brachyistius*, *P. microphthalmus*, *Brevimyrus niger*, and *Pollimyrus adspersus*), and two species with oscillating receptors (*P. tenuicauda* and *P. soudanensis*), plotted head-positive up with normalized peak-to-peak heights. (**B**, **C**) Average power spectra of EODs from species with spiking receptors (**B**) and from species with

*Figure 4. Continued*

oscillating receptors (**C**). (**D**) Frequency tuning curves for representative spiking receptors from four species. Threshold was determined as the lowest intensity stimulus that elicited a spiking response. The frequency with the lowest threshold was taken as the receptor's best frequency. Thresholds were defined as dB relative to the threshold at each receptor's best frequency. Open symbols indicate stimuli for which there was no response from the receptor at the intensity shown, but responses to higher intensities were not recorded. (**E**) Frequency tuning curves for a representative oscillating receptor from *P. tenuicauda* at three intensities. Stimuli were single-cycle bipolar sine waves with positive polarity (peak preceding trough). Vector strength was used as a measure of phase-locking across responses. Vector strength equals 1 when the phase of the oscillatory reset is the same for each stimulus presentation and 0 when the phase of oscillatory reset is completely random for each stimulus presentation. The frequency that elicited the highest vector strength was taken as each receptor's best frequency. (**F**) Average receptor best frequency vs average conspecific EOD peak power frequency for all species studied. Best frequencies were averaged across responses to positive- and negative- (trough preceding peak) polarity stimuli in oscillating receptors. We used the best frequencies at 10 nA in *P. tenuicauda*. Closed symbols denote species with spiking receptors and open symbols denote species with oscillating receptors. (**G**) Vector strength of oscillating responses to positive-polarity single-cycle bipolar sine stimuli at multiples of receptors' spontaneous oscillation periods at three intensities in *P. tenuicauda*. Each point in **F** and **G** represents the average across receptors where appropriate, and error bars represent S.E.M.

In agreement with previous studies (*Bell and Grant, 1989*; *Baker et al., 2013*), spiking receptors encoded IPIs into interspike intervals within receptors, with the interspike intervals exactly matching the IPI (*Figure 5A,D*). IPIs shorter than 1 ms in *Pollimyrus adspersus* and *P. microphthalmus* failed to evoke spikes in response to the second pulse due to receptors' refractory periods (*Figure 5B,C*), which is why data points below these values are missing from *Figure 5D*. Spiking receptors can therefore faithfully encode the IPIs generated by individual signaling fish into interspike intervals within receptors.

## Oscillating receptors encode IPIs into interoscillation intervals and amplitude increases

To understand IPI coding by oscillatory receptors, we analyzed the amplitude and phase of oscillations as a function of the interval between pulse pairs (e.g., *Figure 5E,F*). At the shortest intervals tested, the stimulus artifact of the second pulse obscured the oscillatory response to the first pulse (*Figure 5F*). Thus, to investigate how oscillating receptors encode IPIs, we measured interoscillation intervals as the time between the first oscillatory peak following a single-pulse stimulus and the first oscillatory peak following the second pulse in the paired stimulus (*Figure 5E,F*). This method was equivalent to measuring the time interval between oscillatory peaks evoked by each pulse in the pair, but it allowed us to measure the interoscillation intervals at the shortest IPIs where the oscillatory response was obscured by stimulus artifact. Interoscillation intervals accurately represented the IPI for intervals of ~1 ms and longer (*Figure 5E,G*). Importantly, interoscillation intervals accurately encoded IPIs over the range produced by individual fish ($\geq$8 ms) (*Hopkins, 1986*; *Carlson, 2002*).

Although interoscillation intervals accurately represent IPIs longer than ~1 ms, how would the system disambiguate these stimulus-evoked oscillatory peaks from ongoing oscillations? Phase resets in oscillatory receptors serve to transiently synchronize the population in response to a stimulus (*Figure 6*). Because oscillation frequency varies across receptors, responses rapidly desynchronize (*Figure 6*). For instance, in response to a single square pulse, across-receptor synchrony is greatest for the first poststimulus oscillatory peak (*Figure 6A*). In response to a pair of pulses, across-receptor synchrony is high immediately following each pulse and rapidly decreases (*Figure 6B*). To further illustrate this, we recorded responses to a single conspecific EOD of normal and reversed polarities from all 36 receptors in a single rosette of one *P. tenuicauda* (same receptors illustrated in *Figure 2B*). Responses from four of these receptors are shown in *Figure 6C*. Notice that again, across-receptor synchrony is greatest for the first poststimulus oscillation, but then rapidly declines. Summing responses across all 36 receptors resulted in a single, distinct peak immediately following the stimulus (*Figure 6D*), due to a transient synchronization across receptors followed by increasing asynchrony of receptors oscillating at different frequencies. In this way, the electrosensory system could use

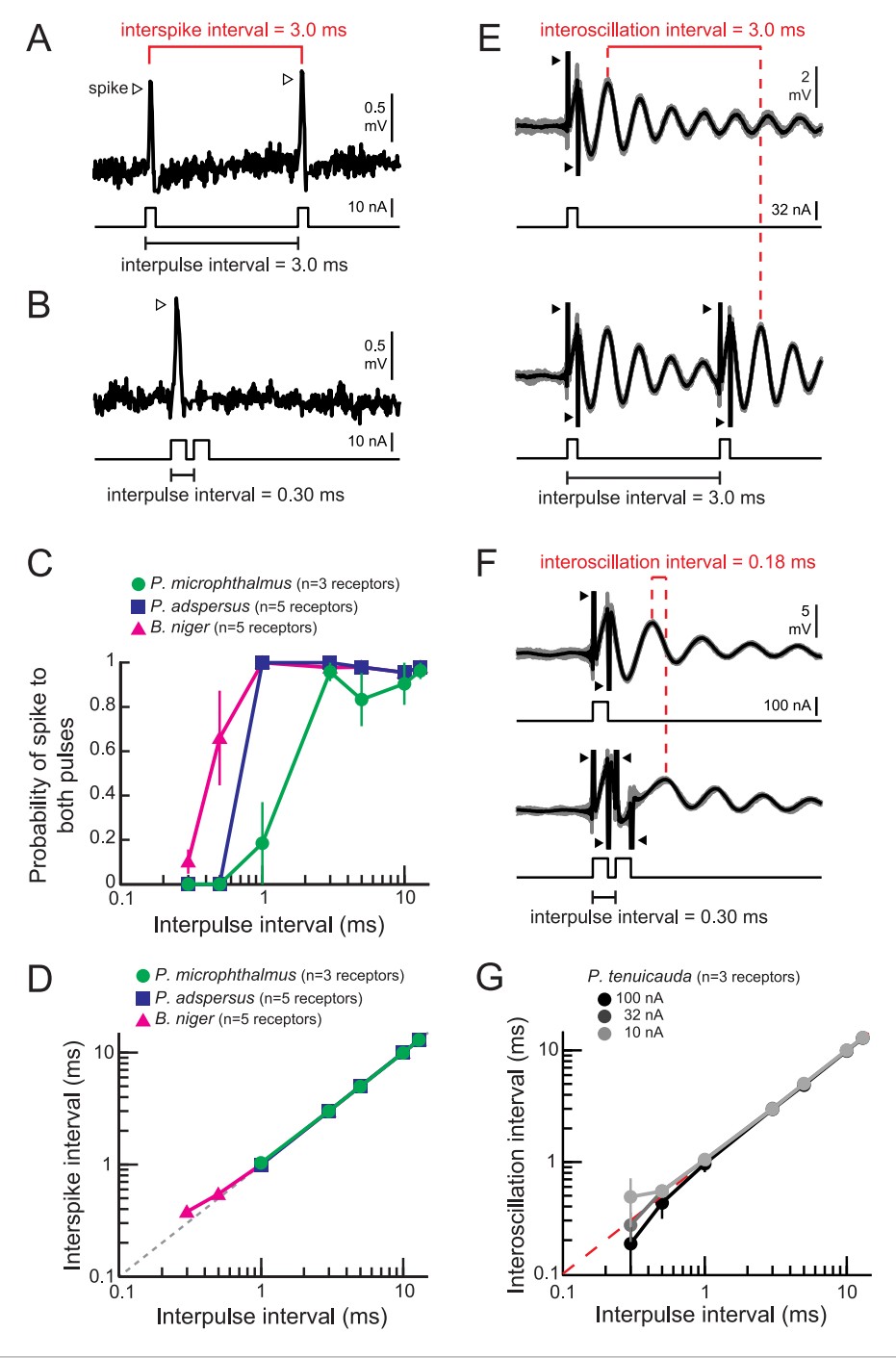

**Figure 5**. Spiking and oscillating receptors encode interpulse intervals into interspike and interoscillation intervals, respectively. (**A**, **B**) Extracellular recording from a spiking receptor in *B. niger* in response to a pair of positive-polarity monopolar square pulses of 0.2-ms duration and 3.0-ms interpulse interval (IPI) (**A**) and 0.30-ms IPI (**B**). A spike occurred in response to the first pulse only for the 0.30-ms IPI. Stimulus artifacts were removed from recordings for clarity. (**C**) The probability that a receptor fired spikes to both positive-polarity pulses in a pair vs IPI for spiking receptors from three species. (**D**) Interspike interval vs positive-polarity IPIs for the same spiking receptors shown in **C**. The receptors of *P. adspersus* and *P. microphthalmus* did not fire spikes in response to both pulses when IPIs were shorter than 1 ms, so there are no data points at these intervals. (**E**) Extracellular recordings from an oscillating receptor in *P. tenuicauda* in response to a single pulse (top) and to a pair of pulses with 3.0-ms IPI (bottom). Responses to each stimulus presentation are shown in gray and the average across stimulus presentations is shown

*Figure 5. continued on next page*

*Figure 5. Continued*
in black. The interoscillation interval was defined as the time interval between the first poststimulus oscillatory peak evoked by the single pulse and that evoked by the second pulse in the pair and was measured from the averaged traces. (**F**) Same as **E** for 0.30-ms IPI. (**G**) Interoscillation interval vs IPI for the responses of *P. tenuicauda* receptors to positive-polarity stimuli. Each point in **C**, **D**, and **G** represents the mean across receptors and error bars represent S.E.M.

synchrony across the population of receptors to distinguish between stimulus-evoked and spontaneous oscillation cycles.

Oscillating receptors also encoded IPI into amplitude changes (*Figure 7A,B*). At the highest intensity tested, oscillation amplitudes were attenuated relative to single-pulse responses for IPIs below 1 ms (black curve in *Figure 7C*). As stimulus intensity decreased, however, oscillation amplitudes became selectively enhanced in response to 0.5-ms IPIs (*Figure 7C*). Phase-locking, as measured by the vector strength of oscillatory responses to the second pulse in the pair, also depended on IPI and stimulus intensity (*Figure 7E*). As the intensity decreased, responses became more sharply selective for IPIs around 0.5 ms, which is near the intrinsic oscillation periods (0.47–0.99 ms) of receptors in this species. This is suggestive of resonance in which stimulating an oscillating receptor with a pair of pulses separated by the receptor's intrinsic oscillation period results in stronger responses.

To test this hypothesis, we presented receptors with IPIs equal to multiples of their intrinsic oscillation periods. As stimulus intensity decreased, oscillatory responses became more selective for IPIs equal to the intrinsic oscillation period (*Figure 7D*). At the lowest intensity tested, the phase-locking was also greatest for IPIs matching receptors' intrinsic oscillation periods (light gray curve in *Figure 7F*).

These results reveal that oscillating receptors produce the largest responses for IPIs matching their intrinsic oscillation periods. These periods, however, are at least one order of magnitude shorter than the IPIs that single fish produce. Therefore, signals arriving at submillisecond IPIs will be easier to detect, even though these IPIs are too short for each individual pulse within the train to be encoded with a phase reset. Why would some species of mormyrids have receptors that are maximally sensitive to signals too short to be produced by individual fish?

## Oscillating receptors respond most strongly to submillisecond IPIs present in communication signals recorded from a group of fish

Even though the minimum IPI a single fish produces is approximately 8 ms, a large group of signaling fish may collectively produce much shorter IPIs. Could the oscillations in rosette receptors be tuned to these shorter IPIs? To test this hypothesis, we first recorded 20 min of electric signaling from individual fish of a species with spiking receptors (*P. microphthalmus*) (e.g., *Figure 8A*) and from individual fish of a congeneric species with oscillating receptors (*P. tenuicauda*) (e.g., *Figure 8D*). Next, we recorded electrical activity from group tanks of each of the same two species (*Figure 8B,E*). Indeed, group signals contained much shorter IPIs than single-fish signals (compare *Figure 8B* with *Figure 8A*, and *Figure 8E* with *Figure 8D*; note different time scales).

To determine whether spiking and oscillating receptors could encode the very short IPIs in group signals, we compared the magnitude of receptor responses with the IPIs recorded from a group of the corresponding species (*Figure 8C,F*). The spiking receptors of *P. microphthalmus* could not respond to the submillisecond intervals present in collective conspecific group signals (*Figure 8C*). In contrast, oscillating receptors responded most strongly to 0.5-ms IPIs, which are found only in group signals (*Figure 8F*). Thus, one advantage for species with high-frequency oscillating receptors may be the ability to detect very short IPIs in group communication signals that spiking receptors cannot encode due to refractoriness.

## Behavioral responses of a species with oscillating receptors reveal tuning to submillisecond IPIs

If oscillating receptors facilitate detection of communication signals produced by a group of individuals, then species with oscillating receptors should exhibit selective behavioral responses to

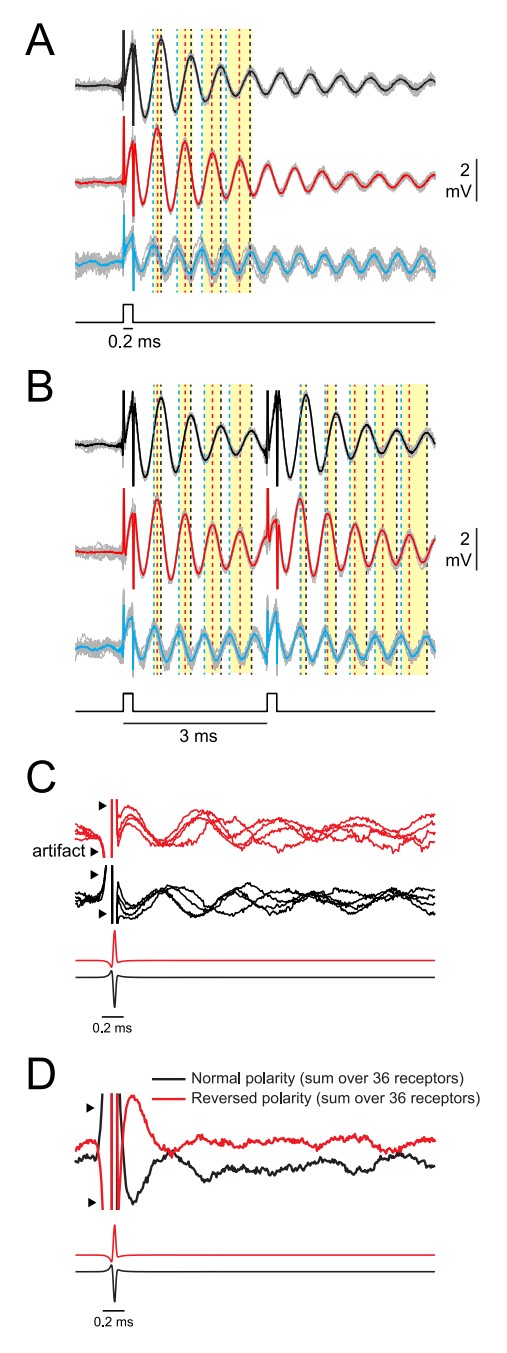

**Figure 6.** Synchrony across receptors is greatest for the first poststimulus oscillation and then rapidly declines. (**A**) Responses of three oscillating receptors in *P. tenuicauda* to a single square pulse delivered in sequential recordings. Responses to each of 10 stimulus presentations are shown in gray and averages are shown in black, red, or blue. Dotted vertical lines in the corresponding color denote the times of the first four poststimulus oscillatory peaks. Yellow bars group the first, second, third, and fourth peaks from each receptor. Note how the peaks are transiently synchronized just after the stimulus, but become increasingly asynchro-
*Figure 6. continued on next page*

submillisecond IPIs, and species with spiking receptors should not. To test this hypothesis, we performed behavioral playback experiments in which we presented a single fish with a train of 10 conspecific EODs at constant IPIs (*Figure 9*).

In general, mormyrids respond to electric signals by increasing or decreasing their rate of EOD production (*Moller et al., 1989*; *Post and von der Emde, 1999*; *Carlson et al., 2011*; *Lyons-Warren et al., 2012*). Increases in EOD rate have been interpreted to be an orienting response to a novel stimulus (*Post and von der Emde, 1999*). Decreases in EOD rate, or social silence, have been hypothesized to allow the silent fish to attend to another fish's signals or to make the silent fish electrically 'invisible' to other fish (*Moller et al., 1989*). As previously described in another species (*Post and von der Emde, 1999*), the same fish often produced both an increase and a decrease in response to a stimulus (*Figure 9*). In general, *P. tenuicauda* tended to respond with a decrease followed by an increase (*Figure 9A*), whereas *P. microphthalmus* tended to respond with an increase followed by a decrease (*Figure 9B*). Regardless of an individual's response pattern, we measured the maximum increase and decrease in EOD rate in response to each stimulus and compared the fish's electric signaling activity evoked by IPI stimuli to that evoked by a single conspecific EOD. A species with oscillating receptors produced the strongest EOD rate increases at the shortest and longest IPIs tested (*P. tenuicauda* in *Figure 9C*). The shortest IPIs ($\leq$5 ms) correspond to those found only in signals recorded from groups of fish.

A congeneric species with spiking receptors (*P. microphthalmus* in *Figure 9D*) responded differently to IPIs than the species with oscillating receptors (repeated-measures ANOVA, interaction between species and IPI, $F_{(6,12)} = 3.4$, $p = 0.0048$). The species with spiking receptors produced the greatest EOD rate increases for IPIs in the intermediate range of those tested, which roughly correspond to the minimum IPI produced by individual fish.

In general, the species with oscillating receptors tended to produce weaker EOD rate decreases in response to the IPI stimuli than to single pulses (negative *y*-values in *Figure 8B*). In contrast, the species with spiking receptors tended to produce stronger EOD rate decreases for IPI stimuli than for single EODs (positive *y*-values in *Figure 9B*). Accordingly, there was a significant effect of species in a repeated-measures ANOVA ($F_{(1)} = 18$, $p = 0.0012$), but no interaction effect between species and IPI ($F_{(6,12)}$

*Figure 6. Continued*

nous with each subsequent cycle. (**B**) Same as **A** for responses of the same three receptors to a 3-ms IPI stimulus. Note the transient increase in synchrony across receptors just after both stimulus pulses. (**C**) A single recording trace from four receptors in the right augenrosette of one *P. tenuicauda* in response to a normal- (black) and reversed- (red) polarity conspecific EOD. Recording traces were normalized to the amplitude of the first poststimulus oscillation. (**D**) The sum of the normalized responses of all 36 receptors in the right augenrosette of one *P. tenuicauda* (illustrated in *Figure 2B*) to a normal- and reversed-polarity conspecific EOD (this includes the four traces shown in **C** as well as responses from the 32 additional receptors). The enhanced synchrony across receptors for the first post-stimulus oscillatory peak results in the largest peak in the summed response just after the stimulus.

= 1.0, p = 0.43). The potential social implications of variation in the strength of EOD rate increases and decreases across this range of IPIs are unknown.

These results lend support to our hypothesis that the frequency tuning of oscillating receptors may mediate detection of submillisecond IPIs that are only present in the communication signals produced by a large group of individuals. These very short IPIs elicit stronger responses in oscillating receptors and greater EOD rate increases in species with oscillatory receptors. Since spiking receptors cannot encode these submillisecond IPIs, we propose that oscillations are an adapation for detecting large groups of conspecifics.

## Discussion

Here, we describe a novel mechanism for peripheral sensory coding in which sensory receptors with spontaneous oscillatory activity respond to stimuli with phase resets that result in transient synchrony across the population. The degree of synchrony provides information about pulse timing and polarity, but it does not reflect the precise timing cues within the pulse waveform, consistent with the inability of these fish to behaviorally detect EOD waveform variation (*Carlson et al., 2011*). Further, oscillating receptors are most sensitive to submillisecond IPIs that match their oscillation period (0.47–0.99 ms). This sensitivity is remarkable for two reasons. First, these submillisecond IPIs are too short for spiking receptors to reliably encode. Second, these IPIs are much shorter than those produced by individual fish, suggesting increased sensitivity to communication signals produced by groups of individuals. Indeed, playback experiments demonstrated that a species with oscillating receptors responded strongly to submillisecond IPIs within the range of enhanced receptor sensitivity. Thus, this novel peripheral coding mechanism may represent an adaptation for detecting communication signals from large groups of fish.

Oscillatory electrical activity is widespread in neural networks (*Buzsaki and Draguhn, 2004*). Many studies have linked phase resets in ongoing cortical oscillations to perceptual performance, suggesting an important role for phase resets in sensory processing (*Thorne and Debener, 2014*). However, whether these phase resets encode specific information about a sensory stimulus or instead serve a modulatory function remains to be determined (*Lakatos et al., 2009*). Futhermore, all phase resets previously described occur in central circuits and involve oscillations of much lower frequencies (<100 Hz) than those (1–3 kHz) in the sensory receptors we describe (*Harder, 1968b*; *Thut et al., 2012*; *Canavier, 2015*). Using in vivo extracellular recordings from single receptors, we demonstrate that phase resets in the periphery lead to transient synchrony among receptors, enabling detection of and entrainment to very fast temporal patterns in communication signals. Sensitivity to temporal patterns has been shown to involve network interactions in a wide range of central circuits (*Klug et al., 2012*; *Buonomano, 2014*). Here, we show that electrical oscillations in sensory receptors can act as peripheral filters for temporal patterns.

Spiking receptors are most sensitive to stimulus frequencies near those in conspecific EODs (*Hopkins, 1981*; *Bass and Hopkins, 1984*; *Lyons-Warren et al., 2012*). Peripheral sensory receptors whose maximal sensitivity corresponds to a principal feature present in the signal they detect are considered 'matched filters' (*Wehner, 1987*). Spiking receptors thus appear to be matched filters for conspecific EODs. Moreover, the precisely timed spikes in these receptors enable discrimination of waveform variation (*Hopkins and Bass, 1981*; *Carlson et al., 2011*), which would facilitate signaler identification. In fact, studies on species with spiking receptors have revealed behavioral sensitivity to individual differences in conspecific EODs (*Graff and Kramer, 1992*; *Hanika and Kramer, 2005*). Collectively, these results support the hypothesis that spiking receptors are specialized for detecting communication signals produced by individual fish and encoding the information necessary to identify the sender.

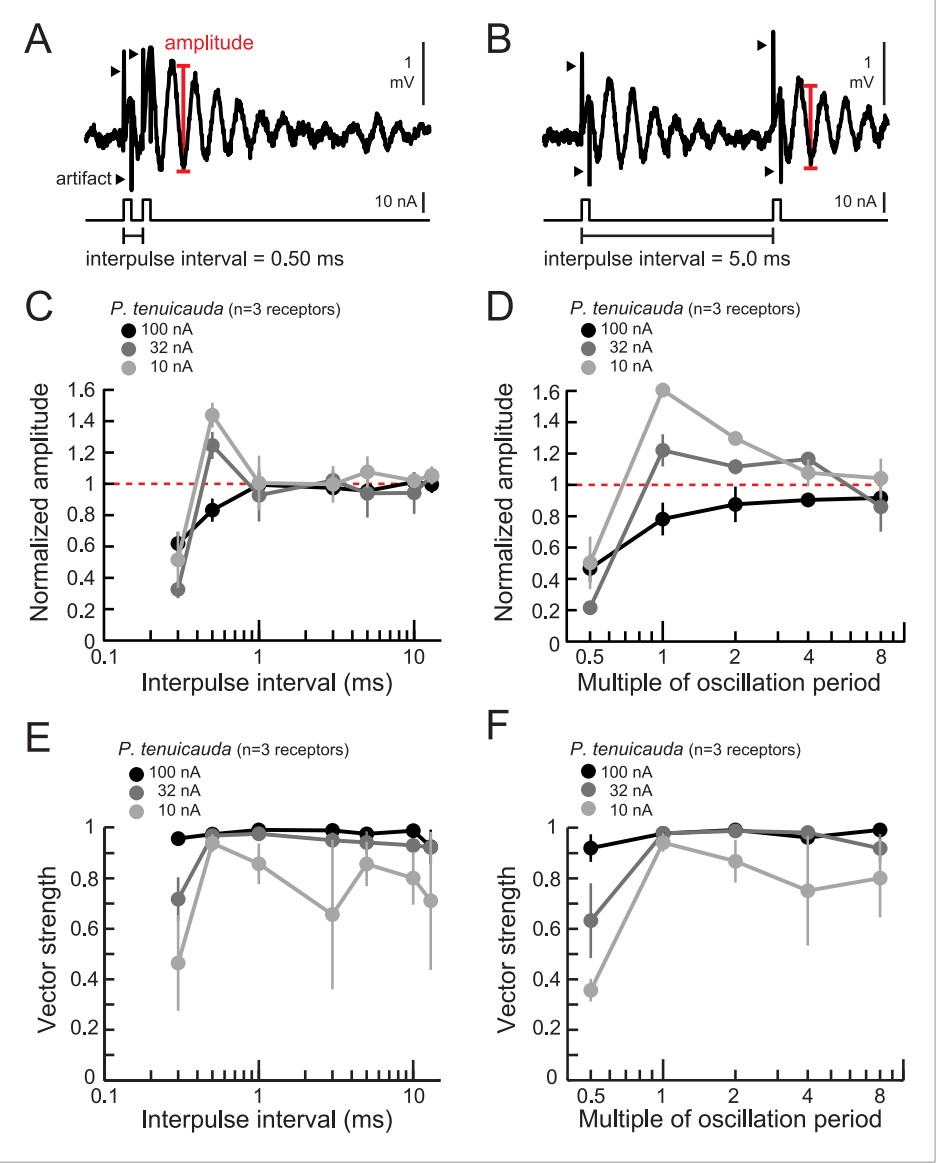

**Figure 7**. Oscillating receptors produce enhanced oscillation amplitudes at submillisecond IPIs matching their intrinsic oscillation periods. (**A**, **B**) Extracellular recording from an oscillating receptor in *P. tenuicauda* in response to a pair of monopolar square pulses of 0.2-ms duration and 0.50-ms IPI (**A**) and 5.0-ms IPI (**B**). We measured the oscillation amplitude on each stimulus presentation as the mean voltage of the first two poststimulus oscillatory peaks minus the voltage at the intervening trough. We then averaged amplitudes across all presentations of the same stimulus. (**C**) Oscillation amplitude evoked by the second pulse in the pair normalized to the amplitude evoked by a single pulse vs IPI for the responses of *P. tenuicauda* receptors at three stimulus intensities. Data shown are for positive-polarity pulses. (**D**) Same as **C** for IPIs corresponding to multiples of oscillating receptors' intrinsic oscillation periods. (**E**) Vector strength vs IPI for oscillating responses to positive-polarity stimuli at three intensities in the same receptors shown in **C** and **D**. (**F**) Same as **E** for IPIs corresponding to multiples of oscillating receptors' intrinsic oscillation periods for the same receptors shown in **C**–**E**. Each point in **C**–**F** represents the mean across receptors and error bars represent S.E.M.

In contrast, oscillating receptors exhibit matched filters not for EOD waveforms, but for IPI patterns in group signals. A receptor's oscillation frequency is predictive of its frequency tuning. However, oscillating receptors are tuned to much lower frequencies than the peak power frequencies of conspecific EODs. Furthermore, these oscillation frequencies confer enhanced sensitivity to IPIs that can only be produced by groups of fish. Therefore, whereas spiking receptors are specialized for

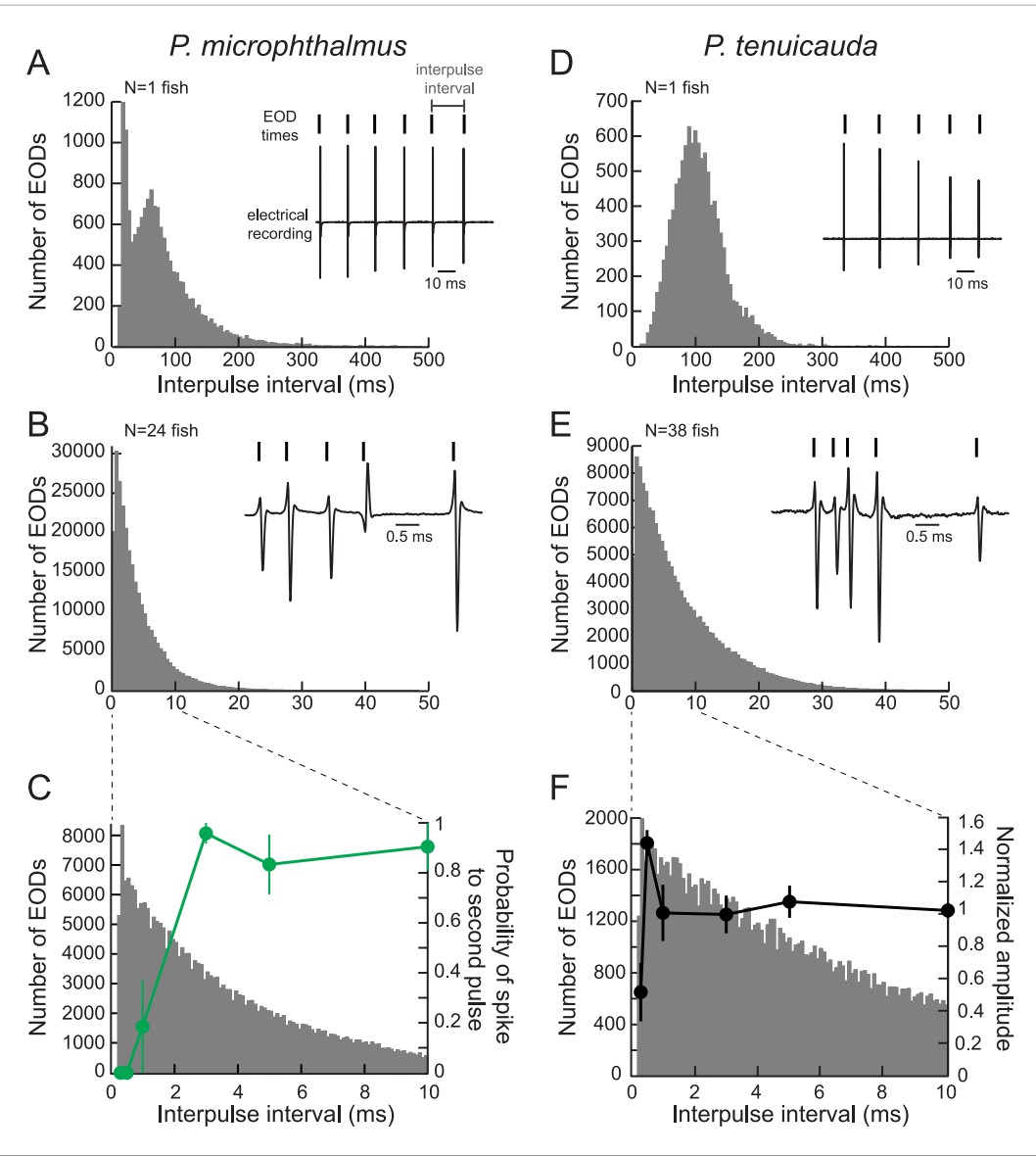

**Figure 8**. Oscillating receptors are most sensitive to submillisecond IPIs occurring in group communication signals. (**A**) A histogram of the IPIs recorded over 20 min from a single fish with spiking receptors (*P. microphthalmus*). Inset, illustration of IPI calculation. We recorded the electric signaling activity from a single fish and recorded the time of each EOD as the time at which the rectified potential crossed a predefined threshold (tick marks above electrical recording trace). We then calculated IPIs as the time between successive EODs. (**B**) Same as **A** for a recording from a group tank of 24 *P. microphthalmus*. Inset, electrical recording from the same group of fish illustrating submillisecond IPIs. EOD polarity and amplitude depend on fish's orientation and location relative to the recording electrode. (**C**) Spike probability of three *P. microphthalmus* receptors (green; *y*-axis on right; same data as in *Figure 5C*) vs IPI superimposed on the IPI histogram from **B**. Note the expanded data range in the *x*-axis. Each point represents the mean across three receptors and error bars represent S.E.M. (**D**, **E**) Same as in **A** and **B** for a congeneric species with oscillating receptors (*P. tenuicauda*). (**F**) Normalized oscillation amplitudes of three *P. tenuicauda* receptors (black; *y*-axis on right; same data as in *Figure 6C* at 10 nA) vs IPI superimposed on the IPI histogram from **E**. Note the expanded data range in the *x*-axis. Each point represents the mean across three receptors and error bars represent S.E.M.

detecting signals from individuals, oscillating receptors appear specialized for detecting signals from a large group. Importantly, the submillisecond IPIs to which oscillating receptors are most sensitive do not necessarily reflect concerted signaling amongst the group. It will be interesting to determine

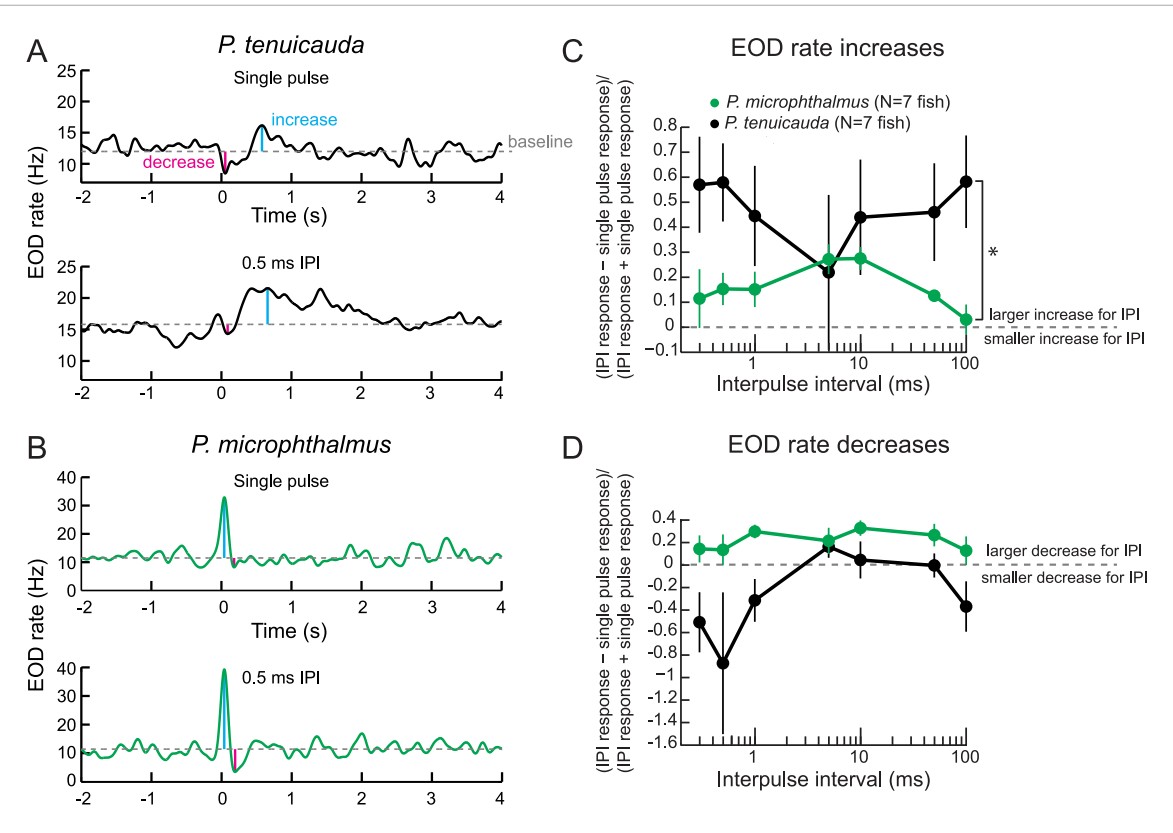

**Figure 9**. Behavioral responses reveal tuning to submillisecond IPIs in a species with oscillating receptors. (**A**) The EOD rate over time in response to a single conspecific EOD (top) and a train of 10 conspecific EODs at constant 0.5-ms IPI in one *P. tenuicauda*. Stimuli were delivered at time = 0 s. We defined the baseline response as the average EOD rate between 4.8 s and 0.2 s before stimulus onset. We measured the maximum increase and decrease in EOD rate relative to baseline that occurred in a window starting 0.2 s before stimulus onset and ending 1.2 s after stimulus offset. (**B**) Same as **A** for one *P. microphthalmus*. (**C**) A normalized measure of EOD rate increases to constant IPI trains of conspecific EODs vs IPI during behavioral playback experiments for a species with oscillating receptors (*P. tenuicauda*) and a congeneric species with spiking receptors (*P. microphthalmus*). *, repeated-measures ANOVA, interaction between species and IPI, p < 0.01. (**D**) Same as **C** for EOD rate decreases. Each point represents the mean across fish and error bars represent S.E.M.

whether fish modulate their IPI patterns in groups and how this might affect the IPI distribution of collective signals.

Differences in receptor physiology and tuning may relate to differences in social behavior. Species in the lineage with oscillatory receptors have been reported to school as adults and prefer open water (*Nichols and Griscom, 1917*; *Hopkins, 1980*, *1981*; *Chapman et al., 1996*; *Lavoue et al., 2004*; *Lavoue, 2012*), unlike other species, which have generally been described as solitary and territorial, and prefer areas with dense vegetation and detritus (*Nichols and Griscom, 1917*; *Hopkins, 1980*, *1981*; *Hopkins and Bass, 1981*; *Chapman et al., 1996*; *Friedman and Hopkins, 1996*; *Lavoue, 2012*). In single-species lab tanks, *Petrocephalus tenuicauda* and *Petrocephalus soudanensis* form open-water shoals and schools, respectively, while *Brevimyrus niger*, *Pollimyrus adspersus*, and *Brienomyrus brachyistius* are solitary and seek shelter (unpub. obs.). Thus, the increased sensitivity of peripheral receptors for communication signals produced by groups of conspecifics may mediate the detection and localization of shoals or schools in fish's native habitats. However, *P. microphthalmus* presents an important exception. These fish, with spiking receptors and associated sensitivity to EOD waveform variation, form schools both in laboratory and natural settings (*Lavoue et al., 2004*). Field studies will be necessary to more fully understand how behavioral and ecological differences between species may have selected for different peripheral coding strategies and the perceptual abilities they confer.

Previous work has demonstrated that electrical oscillations serve as a frequency-tuning mechanism in electroreceptors and non-mammalian hair cells (*Bennett, 1965*, *1967*; *Hopkins, 1976*; *Viancour, 1979*; *Watson and Bastian, 1979*; *Fettiplace and Fuchs, 1999*; *Kawasaki, 2001*, *2005*). Indeed, damped oscillations in spike probability following an electrical stimulus in mormyrid spiking receptors may reflect subthreshold oscillations related to frequency tuning (*Bennett, 1965*, *1967*; *Roth and Szabo, 1972*). Spontaneous, noisy oscillations in paddlefish ampullary electroreceptors are hypothesized to contribute to prey detection (*Neiman and Russell, 2001*, *2011*). In frog saccular hair cells, differences in spontaneous spiking vs oscillatory electrical activity are linked to differences in the density of three types of potassium channels (*Rutherford and Roberts, 2009*). Whether similar mechanisms contribute to high-frequency oscillations in some mormyrid species and spikes in other species remains to be determined. Furthermore, whether the oscillations that can be recorded extracellularly from rosette receptors represent collective spiking activity or non-spiking oscillatory potentials in sensory cells at the base of the receptor pore is also unknown. Importantly, the oscillations in mormyrid rosette receptors are distinct from oscillations in hair cells and other electroreceptors due to their larger amplitudes and higher frequencies, as well as the fact that they respond to stimuli with phase resets. To our knowledge, the phase resets reported here are the first observed in any oscillating sensory receptor.

These stimulus-evoked phase resets result in transient synchrony across oscillating receptors. Since receptors have different intrinsic oscillation frequencies, this synchrony will rapidly decrease. Therefore, pooling the responses of multiple receptors could mediate synchrony detection and allow for the disambiguation of stimulus-evoked and spontaneous oscillations. Whether and where such integration may occur remains to be determined. A single afferent fiber could contact multiple receptors, or multiple afferents may converge onto single postsynaptic neurons in the hindbrain nucleus of the electrosensory lateral line lobe, as reported previously in a species with spiking receptors (*Bell and Grant, 1989*).

The entirety of our knowledge of electric communication signal coding in mormyrids had been limited to species with broadly distributed receptors (see *Baker et al., 2013* for review), which fire spikes in all species that have been studied. These species' behavioral sensitivity to EOD waveform variation is due in part to the precise encoding of waveform timing cues by spiking receptors. Spike-timing differences between receptors on opposite sides of the body are compared in the midbrain anterior exterolateral nucleus (ELa), where excitation and inhibition from different receptive fields establish single-neuron selectivity for stimulus pulse durations (*Friedman and Hopkins, 1998*; *Lyons-Warren et al., 2013*). These duration-selective neurons project to the posterior exterolateral nucleus (ELp), where single-neuron selectivity for IPIs arises (*Carlson, 2009*). Notably, species with rosette receptors have a smaller exterolateral nucleus (EL) that is not subdivided into anterior and posterior regions (*Carlson et al., 2011*). In all three species that have been studied, rosette receptors produce high-frequency oscillations (this study, *Harder, 1968b*). How electric communication signals are processed centrally in the EL of these species remains unknown.

Central processing of oscillatory receptor inputs could contribute to distinguishing self-generated stimuli from external stimuli while maximizing sensitivity to external stimuli. In the absence of an electric stimulus, the spontaneous oscillations across receptors are asynchronized. When the fish emits its own EOD, receptors in all rosettes will experience stimuli of identical polarity based on the direction of current flow through the receptors (*Hopkins, 1986*). This means that oscillations in all receptors will reset to the same phase in synchrony. In this case, subtracting the oscillatory responses of receptors on opposite sides of the body would result in cancellation. In contrast, in the presence of an external electric stimulus, receptors at different locations on the fish's body will experience opposite-polarity stimuli and will reset to opposite phases. Subtraction of these opposite-phase responses would increase oscillation amplitudes relative to single-receptor oscillations, thus potentially increasing sensitivity to external signals.

The electrosensory system could implement a subtraction mechanism if oscillatory receptors on one side of the body mediate excitation, and receptors on the opposite side mediate inhibition onto central neurons. In response to self-generated signals, excitation would arrive coincidentally with inhibition, thus canceling responses. This proposed mechanism would also provide the information necessary to locate external signal sources. Stimuli coming from one direction would elicit maximal excitation from one side of the body and minimal inhibition from the other side, whereas stimuli from the opposite direction would elicit minimal excitation and maximal inhibition. The directionality would presumably be reversed in the contralateral hemisphere, leading to a reversed directional preference.

Given the excitatory–inhibitory interactions found in the ELa of species with broadly distributed receptors (*Friedman and Hopkins, 1998*; *Lyons-Warren et al., 2013*), similar interactions could occur in a homologous circuit within the midbrain EL.

The mormyrid family contains two subfamilies called Mormyrinae and Petrocephalinae (*Sullivan et al., 2000*). Most mormyrine species have broadly distributed receptors, ELa/ELp, and behavioral sensitivity to EOD waveform variation (*Carlson et al., 2011*). In contrast, most petrocephaline species have receptors organized into rosettes as well as an EL midbrain region, and they are not sensitive to EOD waveform variation (*Carlson et al., 2011*). One interesting exception is *P. microphthalmus*, which has broadly distributed receptors and ELa/ELp, and it can also detect EOD waveform variation (*Carlson et al., 2011*). Therefore, *P. microphthalmus* represents a striking case of parallel evolution in which a perceptual ability arose twice independently, once in Mormyrinae and once in *P. microphthalmus* (*Carlson et al., 2011*). Here, we show that the broadly distributed receptors of *P. microphthalmus* fire spikes and display similar physiological properties to the spiking receptors of all mormyrine species studied (this study, *Bennett, 1965*; *Harder, 1968b*; *Hopkins, 1981*; *Arnegard et al., 2006*). Thus, we demonstrate correlated evolutionary changes in peripheral sensory physiology, neuroanatomy, and perception.

In addition to describing the sensory basis for a major perceptual difference among species, our results provide the first illustration of information coding by modulations of ongoing oscillations at the periphery. These results lay the groundwork for future investigations into (1) the cellular mechanisms responsible for generating continuous high-frequency oscillations and phase resets in sensory receptors, (2) cellular and network mechanisms for central processing of oscillatory modulations, and (3) differences in ecology and/or social behavior among species with different peripheral coding strategies.

## Materials and methods

### Animals

15 *B. niger* (standard length [SL] = 5.9–8.6 cm), 17 *P. adspersus* (SL = 6.2–8.6 cm), 20 *P. microphthalmus* (SL = 6.3–8.0 cm), 20 *P. tenuicauda* (SL = 5.5–9.0 cm), and 5 *P. soudanensis* (SL = 8.0–10.0 cm) contributed data to this study. We used fish of both sexes. All fish were acquired through the aquarium trade except for *P. microphthalmus*, which were obtained from Lac Zilé, Gabon. EODs and receptor frequency tuning data from *B. brachyistius* came from a previously published study (*Lyons-Warren et al., 2012*). In the laboratory, we housed fish in conspecific-only group tanks with 12 hr:12 hr light:dark cycle, water conductivity of 200–400 µS/cm, and temperature of 25–29°C. We fed the fish live black worms four times per week. Species of the genus *Petrocephalus* are extremely difficult to obtain. Therefore, to minimize the number of fish used in our physiological recordings, we collected just enough data to detect a robust pattern, which in some cases happened at a low sample size (e.g., n = 3 in *Figures 5, 7*). All procedures were in accordance with the guidelines established by the National Institutes of Health and were approved by the Institutional Animal Care and Use Committee at Washington University in St. Louis.

### Receptor recordings

Extracellular recordings from single receptors were obtained using previously reported methods (*Bennett, 1965*; *Hopkins and Bass, 1981*; *Lyons-Warren et al., 2012*). Briefly, we anesthetized fish in 300 mg/l tricaine methanesulfonate (MS-222, Sigma–Aldrich, St. Louis, MO) and then paralyzed and electrically silenced fish with 20–80 µl of 0.1 mg/ml gallamine triethiodide (Flaxedil, Sigma–Aldrich). We then placed the fish in a 20 × 12.5 × 45 cm chamber filled with freshwater and positioned the fish on a plastic platform with lateral supports. We respirated the fish with freshwater through a pipette tip in the fish's mouth while monitoring the fish's electromotor output using a pair of electrodes placed next to the fish's tail (*Carlson, 2002*). Flaxedil silences the EOD, but external electrodes can record fictive EOD motor commands from spinal electromotor neurons. After the fish had completely recovered from anesthesia, as indicated by the return of fictive EODs, we began the recording session. After receptor recordings, fish were allowed to recover completely from paralysis before being returned to their home tank.

We used electrodes made from borosilicate capillary glass (o.d. = 1 mm, i.d. = 0.5 mm; A-M Systems, Everett, WA, USA). We bent the last ~1 cm of the electrode to a 30° angle and polished the

tip using the flame from a Bunsen burner. We filled the electrode with tank water and placed it in an electrode holder with a Ag-AgCl wire connected to the headstage of the amplifier. We placed the electrode next to, but not quite touching, individual receptors. The fish remained completely under water for the entire recording session. Extracellular activity was referenced to ground, amplified 10 times (Neuroprobe Model 1600, A-M Systems), digitized at a rate of 97.7 kHz (RP2.1, Tucker–Davis Technologies, Alachua, FL, USA), and saved using custom software in Matlab 7.0 (MathWorks, Natick, MA, USA). Spiking responses and some oscillating responses were also low-pass filtered (cut-off frequency = 100 kHz) during recording.

We obtained simultaneous recordings from pairs of oscillating receptors using the same glass electrodes as single-receptor recordings. Extracellular activity was referenced to ground, amplified 100 times, band-pass filtered (0.3–20 kHz) (Model 1800, A-M Systems), digitized at a rate of 97.7 kHz (RP2.1, Tucker–Davis Technologies), and saved in Matlab. We selected one spontaneously oscillating receptor in each rosette, for a total of 14 possible pairs across six rosettes. In one *P. tenuicauda*, we collected five 1-s simultaneous recordings of spontaneous activity in 13 pairs of rosettes. We were unable to see the left and right nackenrosettes at the same time under our microscope in this fish, so we could not obtain simultaneous recordings from this pair of rosettes. In another *P. tenuicauda*, we obtained simultaneous recordings from six pairs of rosettes.

To assess whether receptors within a rosette oscillate at the same frequency, we recorded local field potentials with a differential electrode consisting of a pair of Ag-AgCl wires separated by 5 mm. This method allowed us to record the collective activity of several receptors located closer to one another than separate glass electrodes would allow. Each wire had a diameter of 0.635 mm and uninsulated tip length of 2 mm. We oriented the electrode pair perpendicular to the skin, with the recording electrode next to the skin and reference electrode farther away. Electrical activity was amplified 100 times and band-pass filtered (0.3–5 kHz) (Model 1800, A-M Systems), digitized at a rate of 97.7 kHz (RP2.1, Tucker–Davis Technologies), and saved using custom software in Matlab 7.0.

## Stimulus delivery

All stimuli were generated in Matlab, digital-to-analog converted at a rate of 195.31 kHz (RP2.1, Tucker–Davis Technologies), and attenuated (PA5, Tucker–Davis Technologies) before delivery to the current input of the amplifier (Neuroprobe Model 1600, A-M Systems). The amplifier then delivered constant-current stimuli through the monopolar recording electrode referenced to ground (*Lyons-Warren et al., 2012*). We used a bridge balance to minimize the stimulus artifact.

To study how receptors encode electric pulse waveform, we recorded responses to 10 or 25 repetitions of monopolar square pulses of durations ranging from 0.01 to 5 ms. We used both positive and negative polarities at an intensity within the behaviorally relevant range that reliably elicited responses from each receptor (10–18 nA in *P. adspersus*, 18 nA in *B. niger*, 32 nA in *P. microphthalmus*, and 45–100 nA in *P. tenuicauda*).

We also delivered previously recorded conspecific EOD waveforms. For each receptor, we randomly selected an EOD stimulus from a library of 10 conspecific EOD waveforms. We presented 10 repetitions of both normal- (head-positive) and reversed-polarity waveforms at intensities ranging from 6 to 312 nA.

To measure frequency tuning in spiking receptors, we presented 15 repetitions of 90-ms duration sinusoidal stimuli of frequencies ranging from 0.1 to 50 kHz with 5 ms cosine-squared on- and off-ramps. Because we could not separate a continuously oscillating response from a continuously oscillating stimulus artifact, we had to use different stimuli to measure frequency tuning in oscillating receptors. We presented 10 repetitions of single-cycle bipolar sine wave stimuli of frequencies ranging from 0.2 to 84 kHz to measure frequency tuning of oscillating receptors. We also delivered single-cycle bipolar sine wave stimuli at durations equal to multiples (¼, ½, 1, 2, 4) of each receptor's spontaneous oscillation period. We used both positive (peak leading trough) and negative (trough leading peak) polarities at three stimulus intensities (10, 32, and 100 nA) in *P. tenuicauda*. In *P. soudanensis*, we used positive- and negative-polarity stimuli at 32 nA.

To study how spiking and oscillating receptors encode IPIs in communication signals, we presented 10 repetitions of a pair of monopolar square pulses, with each pulse 0.2 ms in duration. We used IPIs ranging from 0.3 to 13 ms. For oscillatory receptors, we also used IPIs equal to multiples (¼, ½, 1, 2, 4,

8) of each oscillating receptor's intrinsic oscillation period. We used both positive and negative polarities at an intensity that reliably elicited responses in each receptor (6–10 nA in *P. adspersus*, 7–10 nA in *B. niger*, 18 nA in *P. microphthalmus*, and 10–100 nA in *P. tenuicauda*).

## Data analysis

To quantify the responses of spiking receptors, we detected spikes by finding the peak voltage that crossed a manually set threshold specific to each receptor. We measured interspike intervals as the difference in time between consecutive spikes. To measure the responses of oscillating receptors, we first median filtered the recordings using a filter width of 0.1 ms. Next, we detected the first seven oscillatory peaks after stimulus offset on each stimulus sweep, which allowed us to measure the first six poststimulus oscillation periods. Imperfections in artifact balancing had no impact on the detection and measurement of spiking responses. However, sometimes it was unclear whether the first poststimulus oscillatory peak was contaminated by artifact. To assess whether the stimulus artifact affected the first poststimulus oscillation peak, we compared the first poststimulus period to the average of the following five poststimulus periods. If the first period was less than 85% or more than 115% of the average of the next five periods, we concluded that the artifact interfered with the first poststimulus peak and instead used the second poststimulus peak for all measurements.

The time between the first and second poststimulus peaks represented the poststimulus oscillation period. We measured the oscillation amplitude by first averaging the voltage values of the first two poststimulus oscillatory peaks, and then subtracting the voltage value at the intervening oscillatory trough. Next, we calculated the angle ($\varphi$, in radians) of the oscillatory phase reset as

$$\varphi = \frac{lat}{p},$$

where *lat* is the latency of the first poststimulus peak (time of peak minus time of stimulus offset) and *p* is the first poststimulus period. We then used these values across repetitions to calculate the vector strength (*r*), a measure of phase-locking, according to

$$r = \sqrt{\left(\frac{\sum\cos(\varphi)}{n}\right)^2 + \left(\frac{\sum\sin(\varphi)}{n}\right)^2},$$

where *n* is the number of stimulus repetitions. The vector strength is a normalized measure of phase-locking that equals 1 when the constituent angles are perfectly in phase with one another across stimulus repetitions, and 0 when the constituent angles are completely random across stimulus repetitions.

We recorded spontaneous activity of each receptor five times for 1 s each. For a given spiking receptor, we measured the interspike intervals on each repetition and then averaged across recordings. For oscillating receptors, we averaged the fast Fourier transform (FFT) of the oscillatory activity across five repetitions, and then used the peak of the averaged FFT as the receptor's spontaneous oscillation frequency. We used the inverse of the spontaneous oscillation frequency as the spontaneous oscillation period. We measured the spontaneous oscillation amplitude of each receptor by subtracting the voltage at each oscillatory trough from the voltage of the preceding oscillatory peak, and then averaging these amplitude values across the spontaneous recordings. If spontaneous activity was recorded more than once from the same receptor, we averaged measurements across all recordings.

To determine whether the oscillatory activity of different receptors was correlated, we obtained the instantaneous phase of oscillations of two receptors recorded simultaneously using a Hilbert transform in Matlab. We then computed the circular correlation coefficient between the phases of two simultaneous recordings (*Berens, 2009*). For each pair of receptors, we generated two surrogate data sets of non-simultaneous recordings. The first data set consisted of the first 1-s recording from one receptor and the fifth 1-s recording from the other receptor. The second data set consisted of the fifth 1-s recording from one receptor and the first 1-s recording from the other receptor.

We measured frequency tuning in spiking receptors following previously described methods (*Lyons-Warren et al., 2012*). We defined a response as at least one more spike/sweep during the 90-ms stimulus period than during the 90-ms prestimulus period. We determined a receptor's

threshold by finding the lowest intensity at which the receptor responded to a given frequency. The frequency with the lowest threshold was defined as the receptor's best frequency. We generated normalized tuning curves by setting threshold intensities as dB relative to the threshold intensity at the best frequency. For oscillating receptors, we generated tuning curves by plotting vector strength vs stimulus frequency. We defined the best frequency as the frequency with the highest vector strength. In three of eight oscillating receptors, the best frequencies in response to positive- and negative-polarity stimuli differed slightly. In these cases, we determined the receptor's best frequency by averaging the best frequencies to each stimulus polarity.

To study how spiking receptors encode electric pulse waveform, we averaged the time of the first spike that occurred after stimulus onset (positive-polarity stimuli) or stimulus offset (negative-polarity stimuli). The difference in average spike times elicited by stimuli of opposite polarities was defined as the spike-timing difference. Since receptors on opposite sides of the body experience stimuli of opposite polarities, this procedure is equivalent to measuring the difference in spike times between receptors at different locations on the body evoked by the same stimulus (*Hopkins, 1981*). For a comparable measurement in oscillatory responses, we subtracted the average time of the first peak that occurred after the offset of the negative-polarity pulse from the average time of the first peak that occurred after the onset of the positive-polarity pulse for pulse durations longer than 0.5 ms. For shorter stimulus durations (≤0.5 ms), we measured the difference in the times of the first oscillatory peaks that occurred after stimulus offset for each stimulus polarity. This procedure is equivalent to measuring the difference in oscillatory peak times between receptors on opposite sides of the body evoked by the same stimulus. To quantify the degree of oscillation amplitude enhancement, we first measured all prestimulus oscillation amplitudes on each recording trace, and then averaged across all repetitions of the same stimulus. Next, we measured the poststimulus oscillation amplitude on each recording trace and averaged across all repetitions of the same stimulus. Lastly, we divided the mean poststimulus oscillation amplitude by the mean prestimulus oscillation amplitude for each pulse duration.

Since the peak power frequency of a single-cycle bipolar sine wave is slightly lower than the inverse of the wave's duration, we used Welch's power spectral density in Matlab to compute the power spectrum of each single-cycle sine wave. We took the peak of the power spectrum as the frequency of each stimulus.

We measured the difference in the phases of oscillatory resets evoked by opposite-polarity stimuli by first averaging the recorded potential across stimulus repetitions, and then measuring the first poststimulus period and the latency of the first poststimulus peak. We used these values to calculate the difference in the phase ($\Delta\varphi$) of oscillatory reset between responses to opposite-polarity pulses according to

$$\Delta\varphi = \frac{lat_N - lat_P}{\left(\frac{p_N + p_P}{2}\right)} \times 360°,$$

where $lat_N$ is the latency of the first peak, and $p_N$ is the first period, respectively, evoked by the negative-polarity stimulus, and $lat_P$ is the latency of the first peak, and $p_P$ is the first period, respectively, evoked by the positive-polarity stimulus.

To measure spiking responses to paired pulses, we measured the interspike intervals between the spikes elicited by the onset (positive-polarity stimuli) or offset (negative-polarity stimuli) of each pulse. If no spikes occurred in response to the second pulse, no interspike intervals were recorded. For clarity, stimulus artifacts were removed from the recording traces shown in *Figure 5A,B* by drawing a straight line from the times of stimulus onset or offset plus 0.03 ms. In oscillating receptors, the stimulus artifact at the shortest IPIs tested obscured responses to the first pulse in the pair. To circumvent this issue, we first averaged oscillatory responses across stimulus presentations. Next, we measured the time interval between the first oscillatory peak that followed a single pulse and the first oscillatory peak that followed the second pulse in the pair. This method is equivalent to measuring the time interval between the first peaks following each pulse in the pair. We also measured the vector strength and poststimulus oscillation amplitude following the second pulse in the pair. We normalized mean poststimulus oscillation amplitudes by dividing by the mean poststimulus oscillation amplitude following a single pulse of 0.2 ms with the same polarity.

## EOD recording and analysis

EODs were amplified 20–100 times, band-pass filtered (1 Hz–50 kHz) (BMA-200, Odmore, PA, USA), digitized at a rate of 195 kHz (RP2.1, Tucker Davis Technologies), and saved using custom software in Matlab 7.0. We measured total EOD durations using previously defined criteria (*Carlson et al., 2000*). EOD power spectra were computed using Welch's power spectral density estimate in Matlab.

## IPI recordings

To measure the IPIs generated by single fish and groups of fish, we recorded 20 min of electrical activity from a species with spiking receptors (*P. microphthalmus*) and a congeneric species with oscillating receptors (*P. tenuicauda*). For single-fish recordings, fish were removed from their home tank and placed into a 15 × 17 × 31 cm plastic chamber with a single recording electrode in which the positive and negative terminals were separated by 20 cm. Fish were allowed to acclimate for at least 10 min before the recordings. For group recordings, we placed four vertically oriented electrodes suspended with the reference terminal up (16.5 cm from the top of the tank) in the home tank (61 × 76 × 61 cm) of each species. Each electrode was located one-third of the distance between tank walls (i.e., at the vertices of nine equally sized rectangles). We placed the electrodes in the tank and allowed fish to acclimate to them overnight before starting the recordings. We amplified electrical activity 100 times before band-pass filtering (0.1 Hz–20 kHz) (Model 1700, A-M Systems) and digitizing at 97.7 kHz (RX8, Tucker–Davis Technologies). We detected EODs as points where the rectified recorded potential exceeded a threshold of 0.6 mV. Data were saved using custom Matlab software. For the histograms shown in *Figure 8B,C,E,F*, we used the recording from just one of the four electrodes. Single-fish and group tank recordings took place between 1700 and 1900.

## Behavioral playback experiments

To test whether species with different receptor physiologies exhibited different behavioral responses to IPIs, we recorded fish's EOD output in response to a train of 10 conspecific EODs at a constant IPI (0.3–100 ms) using previously described methods (*Carlson et al., 2011*). Briefly, we removed one fish from its group tank and placed it into a 10 gallon tank with a cylindrical plastic chamber (5 cm diameter × 13.2 cm) in the middle of the tank. If the fish was not inside the chamber at the end of a 30-min acclimation period, we guided the fish into the chamber with nets. Netted caps were then placed over each end of the chamber to keep the fish inside during the experiment. Fish were then allowed an additional 5-min acclimation period before starting the experiment. A pair of electrodes oriented horizontally along the inside walls of the chamber delivered the stimulus, and a pair of electrodes oriented vertically at either end of the chamber recorded the fish's EOD output. For each fish, we randomly selected a conspecific EOD waveform stimulus from a library. To generate the IPI train stimuli, we first defined the start and end of the EOD as the first and last points, respectively, where the absolute value of the waveform exceeded 0.5% of the maximum peak-to-peak amplitude. We then concatenated 10 identical EODs together, adding zero padding as necessary in between EODs to achieve the desired IPIs. If the duration of the EOD was longer than the IPI, we truncated the EOD at the IPI. We presented only normal-polarity (head-positive) stimuli. All stimuli were generated in Matlab, digital-to-analog converted at a rate of 195.31 kHz (RX8, Tucker–Davis Technologies), and attenuated (PA5, Tucker–Davis Technologies) before delivery to the analog stimulus isolator (Model 2200, A-M Systems).

To record fish's EOD output, we amplified electrical activity 100 times before band-pass filtering (0.1 Hz–20 kHz) (Model 1700, A-M Systems) and digitizing at 97.7 kHz (RX8, Tucker–Davis Technologies). We collected a fish's responses to 20 repetitions of each IPI train and to a single EOD stimulus, with 1 min between trials to reduce habituation. We randomized stimulus order. To quantify responses, we recorded the time of each EOD produced by the fish, and then computed the spike-density function (SDF) by convolving each EOD time with a Gaussian of 200-ms width (*Carlson and Hopkins, 2004*). We then averaged the SDF across stimulus repetitions. Next, we measured the fish's baseline response by averaging the resulting EOD rate over a window starting 0.2 s after the start of the 5-s prestimulus period and ending 0.2 s before stimulus onset. Because mormyrids can respond to electric stimuli with increases or decreases in their EOD rate (*Moller et al., 1989*; *Post and von der Emde, 1999*; *Carlson et al., 2011*), we measured the maximum and minimum of the SDF that occurred between 0.2 s before stimulus onset and 1.2 s following stimulus offset. We subtracted the baseline discharge rate from these two values to get the maximum increase in EOD rate and the

maximum decrease in EOD rate in response to the stimulus. Because response patterns varied across individuals (*Moller et al., 1989*), we used a normalized measure of response strength. To compare responses across individuals, we subtracted the single-pulse response from the response to the IPI stimulus, and then we divided by the sum of the single-pulse and IPI response. We performed the same analysis on EOD rate increases and decreases.

## Statistics

We used circular statistics to measure mean angles ± S.E.M. of phase modulations of oscillatory receptor responses (*Batschelet, 1981*). We used the parametric second-order Hotelling test for paired circular data to compare oscillatory phase modulations elicited by normal- and reversed-polarity EODs (*Zar, 1999*). We used repeated-measures ANOVA in Statistica 6.1 (StatSoft, Tulsa, OK, USA) to compare behavioral responses to IPI stimuli across two species. We used Wilcoxon's matched-pair test in Statistica to compare circular correlation coefficients of simultaneous and non-simultaneous recordings. We used Spearman's rank test in Statistica 6.1 to compare spontaneous oscillation amplitudes and frequencies, as well as spontaneous and stimulus-evoked oscillation amplitudes in response to square pulses in oscillatory receptors. For all tests, significance level was defined as $\alpha = 0.05$.

All of the data and all of the custom-written Matlab software used for data analysis are available at the Dryad Digital Repository (*Baker et al., 2015*).

## Acknowledgements

We thank Tim Holy and Barani Raman for helpful discussions. This research was supported by the National Science Foundation (IOS-1255396 to BAC) and the National Institutes of Health (DC012452 to CAB).

## Additional information

### Funding

| Funder | Grant reference | Author |
|---|---|---|
| National Science Foundation (NSF) | IOS-1255396 | Bruce A Carlson |
| National Institutes of Health (NIH) | DC012452 | Christa A Baker |

The funders had no role in study design, data collection and interpretation, or the decision to submit the work for publication.

### Author contributions

CAB, BAC, Conception and design, Acquisition of data, Analysis and interpretation of data, Drafting or revising the article; KRH, Acquisition of data, Analysis and interpretation of data

### Ethics

Animal experimentation: All procedures were in strict accordance with the guidelines established by the National Institutes of Health and were approved by the Institutional Animal Care and Use Committee (Animal Welfare Assurance Number: #A-3381-01) at Washington University in St. Louis. The protocol was approved by the Animal Studies Committee at Washington University in St. Louis (Approval Number: 20130265). Every effort was made to minimize pain and stress. No surgical procedures were performed.

## Additional files

### Major dataset

The following dataset was generated:

| Author(s) | Year | Dataset title | Dataset ID and/or URL | Database, license, and accessibility information |
|---|---|---|---|---|
| Baker CA, Huck RK, Carlson BA | 2015 | Data from: Oscillatory phase reset: a novel mechanism for peripheral sensory coding | http://dx.doi.org/10.5061/dryad.ck54h | Available at Dryad Digital Repository under a CC0 Public Domain Dedication. |

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
