## [Decision Letter]

Thank you for submitting your work entitled “Peripheral sensory coding through oscillatory phase reset in weakly electric fish” for peer review at *eLife*. Your submission has been favorably evaluated by Eve Marder (Senior editor) and three reviewers, one of whom is a member of our Board of Reviewing Editors.

The following individuals responsible for the peer review of your submission have agreed to reveal their identity: Ronald L Calabrese (Reviewing editor and peer reviewer) and John Lewis (peer reviewer). A further reviewer remains anonymous.

The reviewers have discussed the reviews with one another, and the Reviewing editor has drafted this decision to help you prepare a revised submission.

Summary:

The authors present a very interesting electrophysiological and behavioral analysis of oscillatory electroreceptors in a group of mormyrid fishes and compare these receptors to spiking receptors found in other mormyrid species. The oscillating receptors encode electric pulses into phase resets that preserve the information necessary to detect and locate signals. Interestingly they are tuned to interpulse interval (IPI) rates that are well above the frequency range of pulses used by individual fish. These short IPIs are found only in recordings from groups of conspecifics and lead the authors to hypothesize that these receptor properties evolved as a means of detecting groups of conspecifics. The authors then test fish with oscillatory receptors and show that unlike those with spiking receptors they respond behaviorally (changes in electric organ discharge rate) to stimuli with the very short IPIs to which their receptors are optimally tuned.

The paper should be of wide interest in the neuroethology community and also to other sensory physiologists and neuroscientists interested in the role of oscillations in brain networks (especially sensory networks) because it serves as an example of how phase resets can carry critical information.

Essential revisions:

1) The expert reviewers were concerned about how the reset coding proposed might work. One expert reviewer wrote “Figure 5 outlines the problem of coding pairs of input pulses. The spiking receptors respond to each input with a spike until the interval (IPI) is too small (they are refractory). In this case, it is clear (because of the nonlinearity of the spiking mechanism) that the timing of each stimulus pulse is represented in the receptor response (a spike). On the other hand, the oscillating receptors respond to the stimulus pulse with strong phase resets – by this I mean that the “new phase” after the pulse is consistent across trials and does not depend on the timing of the stimulus (Figure 3). The authors argue that this reliable, stimulus-evoked phase reset can signal the timing of multiple pulses (IPI). This makes sense, as the cycle reset is somewhat analogous to a spike in the other receptors. However, it is not clear how downstream neurons would distinguish the reset cycles from the others. For example, when the IPI is longer than the cycle period (Figure 5, lower panel), what distinguishes the second cycle (i.e. second peak) following the first stimulus, from the cycle immediately following the second stimulus? Similarly, when the IPI is shorter than the cycle period (Figure 5 lower panel), the time between the cycles following each pulse is longer than the intrinsic cycle period. (Note that the receptor would not have access to the single pulse response for comparison, as in the analysis). In such cases, how will ambiguities between stimulus-perturbed cycles and free-running cycles be dealt with? Perhaps I have missed a key assumption but in the very least, a more detailed explanation of how this coding/decoding works would be appreciated.”

In discussions among the reviewers and the Reviewing editor this was further amplified: “The problem with IPI coding with resets is not trivial. The strong reset does function to synchronize receptors, and this could code for detection of a short IPI (in the same way that electroreceptors detect brief communication signals in wave-type fish, Benda et al., Neuron 2006). However, according to data shown in the present study, synchrony is not so brief, but lasts many cycles of the receptor oscillation. When many fish are around, additional short IPI will likely arrive during this time (while receptors are still synchronized) so how would change in synchrony allow detection of these? In this case, the phase reset itself could carry information but would require a reference phase for comparison.”

The authors should clarify these points in their Discussion.

2) The analyses performed to claim that oscillatory receptors are not correlated (Figure 2) are not substantive enough. The expert reviewer wrote: “For the cross-correlogram (Figure 2), the authors should use surrogate datasets to test whether the data are significant or not. There is also concern about the reports in the text saying that cross-correlation coefficients could reach values as high as 0.8. How can such high values not be significantly different from zero? However, the CCG is probably not the best measure to use here. The authors should also test for phase coupling between oscillators. This can be done by extracting the time-dependent phase of each oscillator (e.g. via Hilbert transform), plotting their joint probability distribution, and comparing it to the product of the individual distributions (again use surrogate datasets to test for significance). I am also unsure as to why the authors use different recording/analysis techniques to test whether oscillators within a rosette are correlated or not. Please be consistent and thorough in justifying your choice of measures for determining whether oscillatory receptors are correlated or not as this will very much determine how results obtained from single receptors will generalize to the population level.”

3) The same referee stressed: “In Figure 3, the authors analyze the phase reset of oscillatory receptors caused by pulses of opposite polarity of different widths. While their results show that the phase resets are always out of phase with each other independent of width, this analysis alone is not enough to prove that no information about pulse width is transmitted by these receptors. For example, have the authors looked at the timecourse of the decay in the damped oscillation following the pulse? Strong statements saying there is no information need to be properly substantiated. The authors could also use information theory. The sentence stating that the amplitude enhancement depends on pulse duration would suggest a coding mechanism. This is really critical for the authors' claim that the fact that species with oscillatory receptors cannot distinguish between pulses with different characteristics is caused by the fact that the peripheral receptors do not transmit this information in the first place.”

4) The data of Figure 8 cause some concern. Behavioral/EOD responses – both EOD rate increases and decreases were observed and described in the panels A and B. Did a single individual exhibit both types of responses? Did some stimuli result in no changes? What was the relative frequency of each type of response over individuals and across species? What conditions or stimuli lead to EOD rate increases and what lead to EOD rate decreases? Is this increase vs. decrease random or is there a pattern in individual fish? What is the behavioral significance of a rate decrease vs. a rate increase? What are the implications of the observation that tuning the effect is not obvious for the rate decreases? The description of this analysis should be clarified/justified e.g. why are the authors choosing a measure of “contrast”?

5) The writing is in general clear but still the paper could be improved for a general audience. The Abstract needs more of a general introduction to the problem and needs more context in terms of adaptation to the natural environment. The authors need to set up background/context for each of their experiments.

[Editors' note: further revisions were requested prior to acceptance, as described below.]

Thank you for resubmitting your work entitled “Peripheral sensory coding through oscillatory phase reset in weakly electric fish” for further consideration at *eLife*. Your revised article has been favorably evaluated by Eve Marder (Senior editor), a Reviewing editor, and two reviewers. The manuscript has been improved but there are some remaining issues that need to be addressed before acceptance, as outlined below:

Please address the concerns of reviewer #2.

*Reviewer #2*:

The authors have done a very nice job addressing my concerns. In particular, the new figure explaining transient synchronization among oscillating receptors and the explanation provided for the behavioral analyses are very much appreciated. However, I feel that additional clarification of the “coding by resets” idea is still necessary.

The title of the paper and much of the discussion suggests that sensory information is coded in phase resets. To me, this means that the phase reset varies with some feature of a sensory stimulus, and that phase (or change in phase) information is passed on to downstream networks as a representation of this stimulus feature. It follows that the phase reset (new phase) should depend on the particular value of the sensory feature. Taken in this context, the data presented suggest that only stimulus polarity can be coded by the phase reset; the phase reset is invariant with the other tested stimulus features, so cannot provide any information about these features. That said, the data show very nicely that transient synchrony among receptors (now shown to be sufficiently brief) provides a means to code for sub-millisecond inter-stimulus intervals, and very importantly, that this is reflected in behavioral responses. So my concern is not with the quality of the data or its significance (that is very clear). I just don't see it as “sensory coding by phase reset” and presenting it as such leads to questions and confusion. Rather, information appears to be coded in the level of synchrony, which is changed by strong resets of the oscillating receptor to a single phase (an implicit requirement of a synchrony code). The work is then more appropriately set in the context of “synchrony coding”, and compared, for example, to the coding strategy described by Benda and colleagues in a related electrosensory system (as noted in the first review).

---

## [Author Response]

*1) The expert reviewers were concerned about how the reset coding proposed might work. One expert reviewer wrote “*Figure 5
*outlines the problem of coding pairs of input pulses. The spiking receptors respond to each input with a spike until the interval (IPI) is too small (they are refractory). In this case, it is clear (because of the nonlinearity of the spiking mechanism) that the timing of each stimulus pulse is represented in the receptor response (a spike). On the other hand, the oscillating receptors respond to the stimulus pulse with strong phase resets – by this I mean that the “new phase” after the pulse is consistent across trials and does not depend on the timing of the stimulus (*Figure 3*). The authors argue that this reliable, stimulus-evoked phase reset can signal the timing of multiple pulses (IPI). This makes sense, as the cycle reset is somewhat analogous to a spike in the other receptors. However, it is not clear how downstream neurons would distinguish the reset cycles from the others. For example, when the IPI is longer than the cycle period (*Figure 5*, lower panel), what distinguishes the second cycle (i.e. second peak) following the first stimulus, from the cycle immediately following the second stimulus? Similarly, when the IPI is shorter than the cycle period (*Figure 5
*lower panel), the time between the cycles following each pulse is longer than the intrinsic cycle period. (Note that the receptor would not have access to the single pulse response for comparison, as in the analysis). In such cases, how will ambiguities between stimulus-perturbed cycles and free-running cycles be dealt with? Perhaps I have missed a key assumption but in the very least, a more detailed explanation of how this coding/decoding works would be appreciated*.*”*

We thank the reviewers for identifying this point of confusion. We agree that the ability to distinguish spontaneous oscillations from stimulus-evoked oscillations is key to understanding sensory coding by the oscillating receptors. We have added a figure (Figure 6) along with text in the Results (subsection “Oscillating receptors encode interpulse intervals into interoscillation intervals and amplitude increases”) and Discussion to describe how phase resets establish transient synchrony in oscillating receptors. In response to a stimulus, the oscillations in receptors on a particular part of the fish’s body will reset to the same phase. Because the receptors oscillate at different frequencies, this synchrony is short-lived (Figure 6). Therefore, the electrosensory system would be able to distinguish ongoing oscillations from stimulus-evoked oscillations by detecting synchronous activity across multiple receptors (Figure 6). Furthermore, the amplitudes of individual receptor responses also transiently increase immediately following a stimulus. The combination of amplitude increases within receptors and transient synchrony across receptors likely contributes to signal detection. We do not yet know where in the electrosensory system this comparison would occur, but we have addressed two likely possibilities in the Discussion: “Whether and where such integration may occur remains to be determined. A single afferent fiber could contact multiple receptors, or multiple afferents may converge onto single postsynaptic neurons in the hindbrain nucleus of the electrosensory lateral line lobe, as reported previously in a species with spiking receptors (Bell 1989).”

*In discussions among the reviewers and the Reviewing editor this was further amplified: “The problem with IPI coding with resets is not trivial. The strong reset does function to synchronize receptors, and this could code for detection of a short IPI (in the same way that electroreceptors detect brief communication signals in wave-type fish, Benda et al., Neuron 2006). However, according to data shown in the present study, synchrony is not so brief, but lasts many cycles of the receptor oscillation. When many fish are around, additional short IPI will likely arrive during this time (while receptors are still synchronized) so how would change in synchrony allow detection of these? In this case, the phase reset itself could carry information but would require a reference phase for comparison*.*”*

*The authors should clarify these points in their Discussion*.

We thank the reviewers for identifying this point of confusion. We have added a figure (Figure 6) along with text in the Results (subsection “Oscillating receptors encode interpulse intervals into interoscillation intervals and amplitude increases”) and Discussion to address across-receptor synchrony. In the previous version of the manuscript, we only illustrated synchrony of a single receptor’s responses to multiple presentations of the same stimulus. However, in reality, the electrosensory system does not have access to this information. A more realistic scenario is comparing the activity of multiple receptors to the same stimulus. When many fish are around, submillisecond IPIs will be easier to detect due to the increased oscillation amplitudes, even if these IPIs are too short for each individual pulse within the pulse train to be precisely encoded by a phase reset. We have added clarification on this point to the Results: “Therefore, signals arriving at submillisecond IPIs will be easier to detect, even though these IPIs are too short for each individual pulse within the train to be encoded with a phase reset.”

*2) The analyses performed to claim that oscillatory receptors are not correlated (*Figure 2*) are not substantive enough. The expert reviewer wrote “For the cross-correlogram (*Figure 2*), the authors should use surrogate datasets to test whether the data are significant or not. There is also concern about the reports in the text saying that cross-correlation coefficients could reach values as high as 0.8. How can such high values not be significantly different from zero? However, the CCG is probably not the best measure to use here*.

We thank the reviewers for their constructive criticism. We agree that the cross-correlation is not the appropriate metric to assess synchrony in this case. Accordingly, we have removed the cross-correlation results from Figure 2 and the Results, as well as the method description from Materials and methods.

*The authors should also test for phase coupling between oscillators. This can be done by extracting the time-dependent phase of each oscillator (e.g. via Hilbert transform), plotting their joint probability distribution, and comparing it to the product of the individual distributions (again use surrogate datasets to test for significance)*.

We thank the reviewers for this excellent suggestion. We have added the requested analysis to Figure 2 (Figure 2). For statistical comparison, we performed a circular correlation on the time-dependent phase of each simultaneous recording to assess the degree of correlation between oscillatory activities of two receptors. We generated a surrogate data set of non-simultaneous recordings consisting of the first of five one-second recordings from one receptor in the pair and the last of five recordings from the other receptor. Across 14 pairs of receptors in one *P. tenuicauda*, we found no difference between the circular correlation coefficient of simultaneous recordings (mean = 0.05 + 0.05) and that of non-simultaneous recordings (mean = 4x10^-4^
+ 7x10^-4^) (Wilcoxon matched-pairs test, *Z*_(14)_=1.7, *p*=0.084). In another *P. tenuicauda*, we recorded from six pairs of receptors and found similar results, with no differences in the circular correlation coefficient for phases of simultaneous (mean = -0.06 + 0.04) and non-simultaneous (mean = 0.001 + 0.001) recordings (Wilcoxon matched-pairs test, *Z*_(6)_=1.8, *p*=0.075). We have added these analyses to Materials and methods (subsections “Data analysis”, fourth paragraph and “Statistics”) and a description to the Results (subsection “Spontaneous oscillations of different receptors are not synchronized”).

*I am also unsure as to why the authors use different recording/analysis techniques to test whether oscillators within a rosette are correlated or not. Please be consistent and thorough in justifying your choice of measures for determining whether oscillatory receptors are correlated or not as this will very much determine how results obtained from single receptors will generalize to the population level*.*”*

We thank the reviewers for identifying this point of confusion. Our primary objective in these recordings was to determine whether different receptors oscillate at the same or different frequency. During paired recordings from receptors in different rosettes, we used glass electrodes placed directly adjacent to an individual receptor to ensure that we were recording the extracellular activity of just that receptor. However, the diameter of the glass electrodes needed for these recordings limited us to recording from receptors that were at least 3 mm apart. Within the center of the rosettes, where spontaneous activity is greatest (Figure 2), the receptors are clustered together at much smaller distances (Figure 2). Therefore, we used a differential electrode consisting of a pair of exposed wires to record field potentials resulting from the combined extracellular activity of multiple units near the center of the rosette. Although this method did not allow us to record from isolated receptors simultaneously, it did allow us to determine whether multiple, adjacent receptors within a rosette oscillate at different frequencies. We have added a clarification of these points to the Results (subsection “Spontaneous oscillations of different receptors are not synchronized) and Materials and methods (subsection “Receptor recordings”).

*3) The same referee stressed: “In*
Figure 3*, the authors analyze the phase reset of oscillatory receptors caused by pulses of opposite polarity of different widths. While their results show that the phase resets are always out of phase with each other independent of width, this analysis alone is not enough to prove that no information about pulse width is transmitted by these receptors. For example, have the authors looked at the timecourse of the decay in the damped oscillation following the pulse? Strong statements saying there is no information need to be properly substantiated. The authors could also use information theory. The sentence stating that the amplitude enhancement depends on pulse duration would suggest a coding mechanism. This is really critical for the authors' claim that the fact that species with oscillatory receptors cannot distinguish between pulses with different characteristics is caused by the fact that the peripheral receptors do not transmit this information in the first place*.*”*

We thank the reviewers for identifying this overstatement on our part. We agree with the reviewers that it is difficult to prove that no information at all about pulse duration is transmitted by oscillating receptors. We have scaled back our claims that oscillating receptors encode no information about pulse duration. Instead, we have clarified that oscillating receptors do not encode the same precise timing cues that allow species with spiking receptors to resolve slight waveform variation (Results: “Therefore, species with oscillating receptors cannot behaviorally detect waveform variation at least in part because the precise timing cues are not encoded at the periphery” and Discussion: “These resets provide information about pulse timing and polarity, but do not preserve the precise timing cues within the pulse waveform, consistent with the inability of these fish to behaviorally detect EOD waveform variation (Carlson 2011)).”

*4) The data of*
Figure 8
*cause some concern. Behavioral/EOD responses – both EOD rate increases and decreases were observed and described in the panels A and B. Did a single individual exhibit both types of responses? Did some stimuli result in no changes? What was the relative frequency of each type of response over individuals and across species? What conditions or stimuli lead to EOD rate increases and what lead to EOD rate decreases? Is this increase vs. decrease random or is there a pattern in individual fish? What is the behavioral significance of a rate decrease vs. a rate increase? What are the implications of the observation that tuning the effect is not obvious for the rate decreases? The description of this analysis should be clarified/justified e.g*. *why are the authors choosing a measure of “contrast”?*

We thank the reviewers for identifying our inadequate description of EOD behavioral responses. Increases in EOD rate have been interpreted to be an orienting response to a novel stimulus (Post and von der Emde 1999). Decreases in EOD rate, or social silence, have been hypothesized to allow the silent fish to attend to another fish’s signals, and/or to allow the silent fish to be electrically “invisible” from another fish (Moller, Serrier and Bowling 1989). As previously described in another species (Post and von der Emde 1999), the same fish often produced both an increase and a decrease in response to a stimulus. We have added panels to Figure 9 to illustrate a typical behavioral response to a single EOD and 0.5 ms IPIs in both test species. In general, *P. tenuicauda* tended to respond with a decrease followed by an increase (Figure 9), whereas *P. microphthalmus* tended to respond with an increase followed by a decrease (Figure 9). Regardless of an individual’s response pattern, we measured the maximum increase and decrease in EOD rate in response to each stimulus to capture possible patterns in both types of response. The potential social implications of variation in the strength of EOD rate increases/decreases over this range of IPIs are unknown, but are an interesting topic for future investigation. We used a method of contrast to quantify fish’s responses since individuals can vary in the magnitude of their responses across stimuli, and this served to normalize inter-individual variation. We have added clarification of these points in the Results (subsection “Behavioral responses of a species with oscillating receptors reveal tuning to submillisecond IPIs”) and Materials and methods (subsection “Behavioral playback experiments”).

*5) The writing is in general clear but still the paper could be improved for a general audience. The Abstract needs more of a general introduction to the problem and needs more context in terms of adaptation to the natural environment*.

We thank the reviewers for their constructive criticism. We have made the requested changes to the Abstract.

*The authors need to set up background/context for each of their experiments*.

We thank the reviewers for their constructive criticism. We have included background for each experiment described throughout the Results section.

[Editors' note: further revisions were requested prior to acceptance, as described below.]

Reviewer #2:

*The authors have done a very nice job addressing my concerns. In particular, the new figure explaining transient synchronization among oscillating receptors and the explanation provided for the behavioral analyses are very much appreciated. However, I feel that additional clarification of the “coding by resets” idea is still necessary*.

*The title of the paper and much of the discussion suggests that sensory information is coded in phase resets. To me, this means that the phase reset varies with some feature of a sensory stimulus, and that phase (or change in phase) information is passed on to downstream networks as a representation of this stimulus feature. It follows that the phase reset (new phase) should depend on the particular value of the sensory feature. Taken in this context, the data presented suggest that only stimulus polarity can be coded by the phase reset; the phase reset is invariant with the other tested stimulus features, so cannot provide any information about these features. That said, the data show very nicely that transient synchrony among receptors (now shown to be sufficiently brief) provides a means to code for sub-millisecond inter-stimulus intervals, and very importantly, that this is reflected in behavioral responses. So my concern is not with the quality of the data or its significance (that is very clear). I just don't see it as “sensory coding by phase reset” and presenting it as such leads to questions and confusion. Rather, information appears to be coded in the level of synchrony, which is changed by strong resets of the oscillating receptor to a single phase (an implicit requirement of a synchrony code). The work is then more appropriately set in the context of “synchrony coding”, and compared, for example, to the coding strategy described by Benda and colleagues in a related electrosensory system (as noted in the first review)*.

We agree completely. We have therefore changed the title from “Peripheral sensory coding through oscillatory phase reset in weakly electric fish” to “Peripheral sensory coding through oscillatory synchrony in weakly electric fish”, and we have edited the impact statement accordingly. The Abstract, Introduction and Discussion have been edited accordingly. An additional paragraph in the Discussion has been moved and edited to the same end (seventh paragraph), a minor edit was made to eliminate the resulting repetition (sixth paragraph), and a minor edit has been made to the legend for Figure 2 to clarify that spontaneous oscillations are not synchronized.